# Targeting fatty acid oxidation enhances response to HER2-targeted therapy

Ipshita Nandi[1,2], Linjia Ji[1], Harvey W. Smith[1], Daina Avizonis [1],
Vasilios Papavasiliou[1,2], Cynthia Lavoie[1], Alain Pacis[1,3], Sherif Attalla[1,2],
Virginie Sanguin-Gendreau[1,2] & William J. Muller [1,2] ✉

Metabolic reprogramming, a hallmark of tumorigenesis, involves alterations in glucose and fatty acid metabolism. Here, we investigate the role of Carnitine palmitoyl transferase 1a (Cpt1a), a key enzyme in long-chain fatty acid (LCFA) oxidation, in ErbB-driven breast cancers. In ErbB2+ breast cancer models, ablation of Cpt1a delays tumor onset, growth, and metastasis. However, Cpt1a-deficient cells exhibit increased glucose dependency that enables survival and eventual tumor progression. Consequently, these cells exhibit heightened oxidative stress and upregulated nuclear factor erythroid 2-related factor 2 (Nrf2) activity. Inhibiting Nrf2 or silencing its expression reduces proliferation and glucose consumption in Cpt1a-deficient cells. Combining the ketogenic diet, composed of LCFAs, or an anti-ErbB2 monoclonal antibody (mAb) with Cpt1a deficiency significantly perturbs tumor growth, enhances apoptosis, and reduces lung metastasis. Using an immunocompetent model, we show that Cpt1a inhibition promotes an antitumor immune microenvironment, thereby enhancing the efficacy of anti-ErbB2 mAbs. Our findings underscore the importance of targeting fatty acid oxidation alongside HER2-targeted therapies to combat resistance in HER2+ breast cancer patients.

Tumorigenesis is dependent on the reprogramming of cellular metabolism as both a direct and an indirect consequence of oncogenic mutations. A defining characteristic of cancer cell metabolism is the ability to acquire nutrients from frequently nutrient-poor environments and utilize these nutrients to sustain viability and support biomass generation[1]. A markedly increased consumption of glucose by tumors in comparison to non-proliferating cells was first described several decades ago by Otto Warburg[2]. This observation has been confirmed in various cancers and often correlates with poor patient survival[3]. Glutamine, another principal growth-supporting substrate, contributes not only carbon, but also reduced nitrogen for the de novo biosynthesis of nitrogen-containing compounds such as nucleotides and non-essential amino acids[4]. The heightened demand for glutamine of proliferating tumor cells was initially described by Harry Eagle in the 1950s, who demonstrated that HeLa cells required 10- to 100-fold molar excess of glutamine in culture relative to other amino acids[5].

Glutamine depletion from the tumor microenvironment has been associated with various tumorigenic contexts[6,7]. Altered lipid metabolism, including elevated lipogenesis and fatty acid oxidation (FAO), also plays a significant role in breast cancer cells, particularly considering their proximity to adipocytes that create a fatty-acid rich environment[8,9]. In this context, fatty acids (FAs) are not only consumed for energy production but also act as extrinsic stimuli for cellular growth[10–12].

Carnitine palmitoyl transferase 1 (CPT) is the rate-limiting enzyme facilitating the mitochondrial uptake of long-chain FAs (LCFAs) for FAO[13]. Three isoforms of CPT1 have been characterized, CPT1A, CPT1B and CPT1C[14]. While CPT1B is expressed in skeletal muscles, heart, and adipose tissue, and CPT1C is expressed primarily in the brain, CPT1A has a more widespread pattern of expression[15,16]. Elevated expression of CPT1A is associated with genetic mutations, metabolic disorders, and several cancers, including breast cancer[17]. Although high CPT1A

[1]Rosalind and Morris Goodman Cancer Institute, McGill University, Montreal, QC, Canada. [2]Department of Biochemistry, McGill University, Montreal, QC, Canada. [3]Canadian Centre for Computational Genomics, McGill University, Montreal, QC, Canada. ✉e-mail: william.muller@mcgill.ca

expression is correlates with poor outcome in breast cancer patients, the molecular mechanism underlying this correlation remains elusive[17,18].

The HER/ErbB (Human Epidermal Growth Factor Receptor) family is a class of receptor tyrosine kinases expressed in various tissues of epithelial, mesenchymal, and neuronal origin[19]. These receptors play a crucial role in regulating diverse biological processes including proliferation, differentiation, migration and apoptosis[19]. HER2 is a member of this class of proteins for which, to date, no ligands have been identified[19]. Importantly, the HER2 oncogene is amplified in in 20-30% of breast cancers where it plays a vital part in the development and progression of the disease[19]. Women diagnosed with HER2+ breast cancer are candidates for monoclonal antibody (mAb) therapies that interfere with HER2 by binding to its extracellular domains, including Trastuzumab (Herceptin) and Pertuzumab (Perjeta), as well as tyrosine kinase inhibitors (TKIs) such as Lapatinib, Neratinib and Tucatinib[20-23]. Despite the availability of HER2-targeted therapies, a considerable proportion of HER2+ breast cancers exhibit an aggressive phenotype and eventually relapse due to acquired or intrinsic resistance that enables escape from HER2 inhibition[21,24]. HER2+ breast cancer cells may rewire lipid metabolism, enhancing their uptake, de novo synthesis, storage and oxidation of lipids to fuel growth and resist anti-HER2 treatments[25-27]. Thus, understanding these alternative metabolic pathways is crucial for developing effective therapies for HER2+ breast cancer patients whose disease has progressed beyond resistance to first-line therapies.

Using an ErbB2+ Genetically Engineered Mouse Model (GEMM), we show that Cpt1a ablation delays tumor onset in ErbB2+ tumors, reducing both angiogenesis and metastatic capacity. When established in vitro, Cpt1a-deficient ErbB2+ cells exhibit decreased proliferation and mitochondrial function. Loss of FAO diminishes the generation of electron carriers, NADH and FADH$_2$, inhibiting oxidative phosphorylation (OXPHOS). However, Cpt1a-null cells display a metabolic shift towards glucose metabolism, relying on the TCA cycle for NAD + /FAD reduction. This metabolic adaptation prompted us to explore the ketogenic diet as a potential therapy. This high-fat, low-carbohydrate diet induces the production of ketone bodies as a non-fermentable energy source in the absence of glucose, and has been proposed as an adjuvant therapy for various cancers including breast cancer and glioblastoma[28,29]. Here, we demonstrate that combining Cpt1a inhibition with either the ketogenic diet, rich in LCFAs, or an anti-ErbB2 mAb reduces tumor growth, prolongs survival and enhances anti-tumor immune infiltration. Taken together, these findings propose a promising combination therapy strategy for aggressive HER2+ breast cancers, including those resistant to standard therapy, that integrates interventions targeting signaling and metabolic networks.

## Results

### Cpt1a ablation impairs ErbB2-driven mammary tumor progression and metastasis

Mitochondrial FAO is the major catabolic pathway for fatty acids and plays an essential role in maintaining energy homeostasis[8,9]. As the rate-limiting enzyme of FAO, Cpt1a mediates the transfer of fatty acids into the mitochondria for oxidation. Elevated expression of Cpt1a is associated with genetic mutations, metabolic disorders, and cancers, such as breast cancer due to its lipid-rich environment[17,18,30-32]. To directly address the question of whether reprogramming FAO plays a role in ErbB2/HER2-driven breast cancer progression, we established a unique GEMM combining conditional *Cpt1a* gene targeting (Cpt1a$^{L/L}$) with mammary epithelial co-expression of oncogenic ErbB2 and Cre recombinase (referred to as NIC) (Fig. 1a)[33]. Ablation of both *Cpt1a* alleles (Cpt1a$^{L/L}$) significantly delayed mammary tumorigenesis and severely impaired tumor growth, correlating with reduced proliferation (Fig. 1b–d, Supplementary Fig. 1a). Cpt1a-deficient tumors retained the solid adenocarcinoma pathology typically associated with

ErbB2-expressing GEMMs but showed histological evidence of necrosis and slightly increased apoptosis (Fig. 1e, Supplementary Fig. 1b) as compared to their wild-type (Cpt1a$^{+/+}$) counterparts, as well as decreased vascularity at the periphery, indicating inadequate blood vessel formation (Supplementary Fig. 1c). This suggests that the observed necrosis may be a consequence of decreased blood supply and subsequent nutrient deprivation within the tumor mass[34].

A clinically relevant feature of NIC-derived tumors is their capacity to metastasize to the lungs with high efficiency[35]. Since complete ablation of Cpt1a reduced tumor focality (Supplementary Fig. 1d, e) and proliferative capacity, we examined lung metastasis in mice with equivalent tumor volume to control for potential confounding effects. Tumors deficient in Cpt1a exhibited impaired metastatic capacity as defined by both the number and size of lesions (Fig. 1f, g). In vitro genetic ablation of Cpt1a severely impaired NIC cell migration and invasion, as well as colonization of the lungs in vivo, suggesting further that the observed impact on metastasis was independent of differences in tumor burden (Supplementary Fig. 1f).

### Cpt1a deficiency suppresses FAO, energy metabolism, and tumor cell growth

Tumor cells derived from Cpt1a-deficient tumors proliferated at a significantly lower rate than Cpt1a-proficient cell lines in culture, arguing that the effects of Cpt1a ablation on growth are tumor cell-intrinsic (Fig. 2a). Etomoxir (Etom), which inhibits Cpt1[36], markedly reduced the growth of wild-type NIC cells to levels near those of Cpt1a-deficient cells (Fig. 2b, and Supplementary Fig. 2a) but had no effect on residual proliferation in Cpt1a-deficient cells. Given that Etomoxir inhibits all Cpt1 isoforms, these results indicate that Cpt1a is the primary isoform driving FAO in ErbB2+ tumor cells.

As Cpt1a is modulated by multiple metabolic inputs and implicated in metabolic regulation, we examined the bioenergetic state of Cpt1a-deficient tumor cells[37]. Cpt1a ablation significantly decreased the basal, ATP synthesis-coupled and maximal oxygen consumption rates (OCR) of ErbB2+ tumor cells, indicating an overall suppression of respiration (Fig. 2c and Supplementary Fig. 2b). Loss of Cpt1a also diminished the extracellular acidification rate (ECAR), which was corroborated by decreased lactate secretion (Fig. 2c, d, and Supplementary Fig. 2b). These findings indicate that Cpt1a deficiency reduces OXPHOS and lactate fermentation, consistent with the observed impairment in breast tumor cell proliferation.

The ability to oxidize exogenously added fatty acids (FAs) can be detected by an increase in OCR and is influenced by ATP demand[38]. Exogenously added FAs can also uncouple mitochondria and cause an increase in the OCR independently of FAO[38]. Using a well-established assay to modulate FAO and monitor the utilization of exogenous FAs, we showed that Cpt1a-deficient NIC cells were unable to utilize exogenous palmitate, providing further evidence that Cpt1b and Cpt1c are unable to compensate for the loss of Cpt1a (Fig. 2e and Supplementary Fig. 2c)[38,39]. Accordingly, Cpt1a-deficient NIC cells were unable to oxidize stable-isotope labeled U-$^{13}$C$_6$-Palmitate into Acetyl CoA and the subsequent products of the TCA cycle (Fig. 2f, g). The inability of Cpt1a-deficient tumor cells to metabolize exogenous palmitate could also be attributed to a reduction in FA uptake and the expression of Cd36, an integral plasma membrane protein that imports FAs into cells and is implicated in breast cancer[25,40]. To examine this directly, we employed BODIPY 493/503 and the fluorescently labeled fatty acid analog, BODIPY FL-C16, to quantify lipid uptake and storage in Cpt1a-proficient and -deficient NIC cells[41]. We observed a notable reduction of both BODIPY 493/503 and BODIPY FL-C16 staining in Cpt1a-deficient cells compared to wild-type ErbB2+ cells (Supplementary Fig. 2d–f), indicating diminished uptake and storage of fatty acids in the absence of Cpt1a. In line with this observation, we further demonstrate that Cpt1a-deficient tumor cells express significantly lower levels of Cd36 in vivo, contributing to diminished uptake and utilization of exogenous

lipids (Fig. 2h). Taken together, these data confirm that ErbB2+ breast cancer cells rely on the uptake and oxidation of FAs for energy production and argue that, of the three isoforms of Cpt1, Cpt1a is primarily responsible for regulating FAO in this context.

## Loss of Cpt1a induces metabolic reprogramming in ErbB2+ breast cancer cells

To further explore mechanisms involved in the altered glucose metabolism of Cpt1a-deficient cells, we first measured glucose uptake and the expression of glucose transporters in wild-type and Cpt1a-null ErbB2-driven tumor cells. Both glucose uptake and the expression of the glucose transporter *Glut1* (*Slc2a1*), but not that of *Glut2* (*Slc2a2*), *Glut3* (*Slc2a3*), or *Glut4* (*Slc2a4*), were elevated in Cpt1a-deficient ErbB2 tumor cells (Fig. 3a–c and Supplementary Fig. 3a, b). While Cpt1a loss decreased the conversion of U-$^{13}$C-glucose into lactate, it increased the

labeling of metabolites produced from glycolysis (phosphoenolpyruvate, pyruvate, and alanine) and intermediates of the TCA cycle (citrate, succinate, and malate) (Fig. 3d and Supplementary Fig. 3c, d). This is consistent with increased usage of glucose-derived carbon to fuel the TCA cycle at the expense of lactate fermentation and other metabolic fates of glucose. Accordingly, we observed elevated expression of the mitochondrial pyruvate carrier (Mpc) in Cpt1a-deficient cells (Supplementary Fig. 4a), consistent with increased pyruvate transport into the mitochondria, potentially fueling the TCA cycle to support cell survival[42]. To assess potential dependencies, we silenced the expression of Mpc2, one of the two essential subunits of the Mpc, and utilized the MPC inhibitor, UK-5099. We observed that simultaneous ablation of Cpt1a with Mpc2 silencing or inhibition significantly inhibited proliferation, leading to synthetic lethality (Supplementary Fig. 4b–e). These results underscore the reliance on Mpc

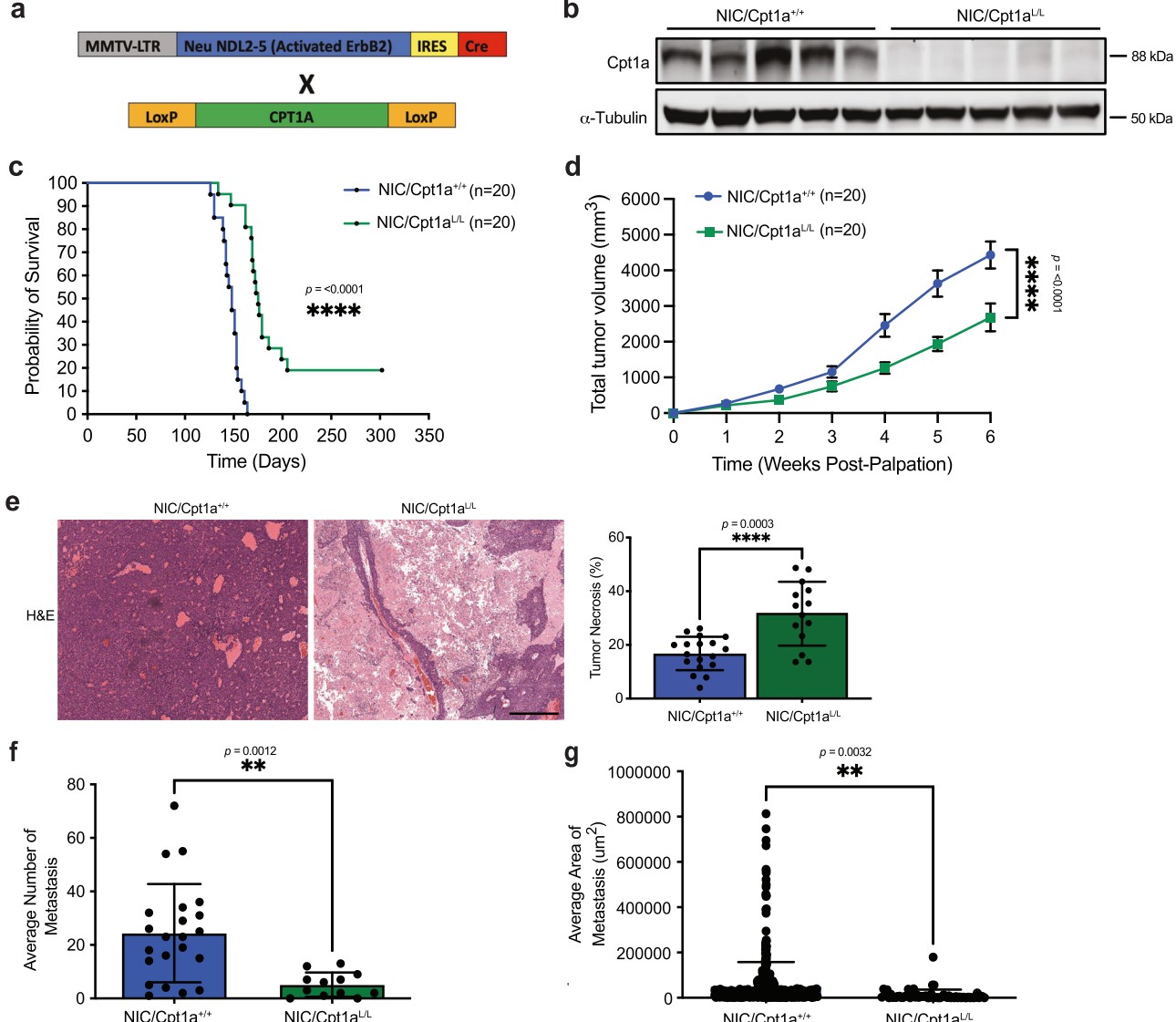

**Fig. 1 | Cpt1a ablation impairs ErbB2-driven mammary tumor progression and metastasis. a** Schematic of genetically engineered mouse model. **b** Lysates from end-stage NIC/Cpt1a$^{+/+}$ and NIC/Cpt1a$^{L/L}$ mice were immunoblotted with antibodies against Cpt1a and α-Tubulin. *n* = 5 mice per genotype in triplicate. **c** Kaplan–Meier analysis of mammary tumor onset in mice with wild-type *Cpt1a* alleles (NIC/Cpt1a$^{+/+}$, *n* = 20) and homozygous (NIC/Cpt1a$^{L/L}$, *n* = 20) conditional *Cpt1a* alleles. ****$p$ < 0.0001; log rank test. **d** Total tumor volume for mice as in (**a**). *n* = 20 mice per genotype - ****$p$ < 0.0001; unpaired, two-tailed Student's *t*-test. **e** Left panel -

Hematoxylin and eosin (H&E) staining of NIC/Cpt1a$^{+/+}$ and NIC/Cpt1a$^{L/L}$ tumors. Images are representative of ten independent tumors of each genotype. Scale bar represents 100 μm. Right Panel – Quantification of the tumor necrosis area as a percentage of the total tumor area. *n* = 15 mice per genotype - *$p$ < 0.05, ****$p$ < 0.0001; unpaired, two-tailed Student's *t*-test. Incidence (**f**) and average burden of lung metastasis (**g**). *n* = 15 mice per genotype - **$p$ < 0.01; unpaired, two-tailed Student's *t*-test. All error bars are expressed as mean values ± SD. Source data are provided as a Source Data file.

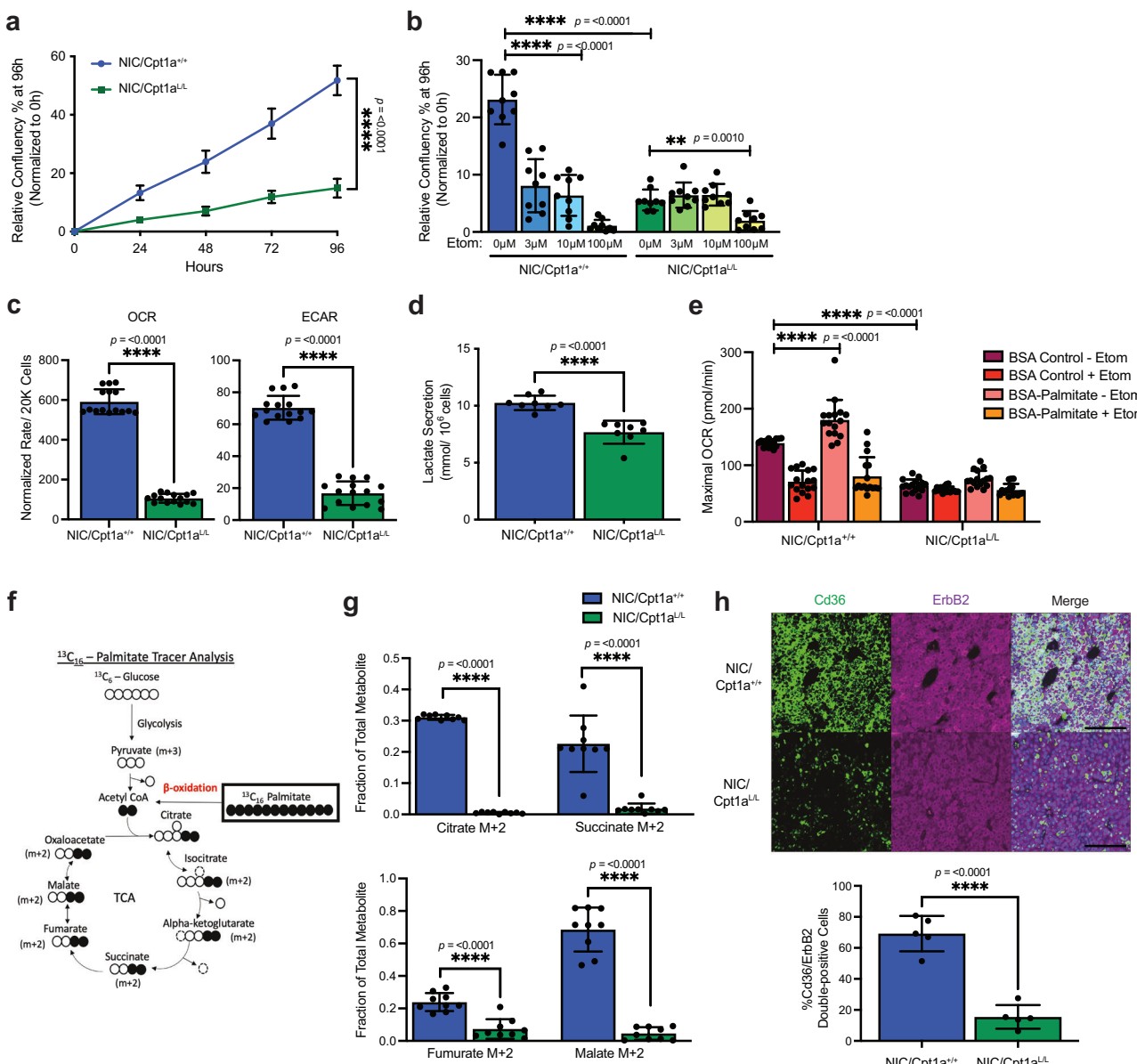

**Fig. 2 | Cpt1a-deficient cells exhibit metabolic dysfunction with reduced proliferation, OXPHOS, glycolysis and FAO. a** Proliferation was assayed in NIC/Cpt1a[+/+] and NIC/Cpt1a[L/L] cells by Incucyte for 96 h. Data were normalized to confluency at $t = 0$. $n = 3$ cell lines in triplicate. $****p < 0.0001$; unpaired, two-tailed Student's $t$-test. **b** Cells in (**a**) were treated with Cpt1a inhibitor, Etomoxir (Etom), at the indicated concentrations or with dimethylsulfoxide (DMSO), and proliferation was assayed using Incucyte. $n = 3$ cell lines in triplicate - $**p < 0.01$ and $****p < 0.0001$; one-way ANOVA with Tukey's post-hoc test. **c** Quantification of basal oxygen consumption rates (OCR) and extracellular acidification rates (ECAR) of NIC/Cpt1a[+/+] and NIC/Cpt1a[L/L] cell lines. $n = 4$ cell lines per genotype in triplicate – $****p < 0.0001$; unpaired, two-tailed Student's $t$-test. **d** Lactate levels in conditioned media from NIC/Cpt1a[+/+] and NIC/Cpt1a[L/L] cells ($n = 4$ cell lines per genotype in duplicate – $***p < 0.001$ and $****p < 0.0001$; unpaired, two-tailed Student's $t$-test). **e** Quantification of maximal OCR in the presence of bovine serum albumin (BSA)-control or palmitate and in the presence or absence of Etomoxir (Etom). $n = 4$ cell lines in quadruplicate - $****p < 0.0001$; one-way ANOVA with Tukey's post-hoc test. **f** Schematic illustrating palmitic acid carbon tracing in the tricarboxylic acid (TCA) cycle. Shown is the pathway of U-[13]C-palmitate carbons (black) transferred among the molecules of the TCA cycle. Palmitic acid is degraded to Acetyl-CoA (M + 2) via fatty acid oxidation, which reacts with oxaloacetate to form citrate (M + 2). **g** U-[13]C-palmitate labeling of TCA cycle M + 2 isotopomers as a fraction of total metabolites. Cells were isolated 24 h after U-[13]C-palmitate tracer. $n = 3$ cell lines in triplicate. $****p < 0.0001$; unpaired, two-tailed Student's $t$-test. **h** NIC tumors were immunostained with the indicated antibodies and DAPI. Right panel - Images representative of ten independent mice per genotype. Scale bar represents 100 µm. Left panel - Staining was quantified and normalized to total cell number (DAPI). Minimum 10,000 cells per sample, $****p < 0.0001$; unpaired, two-tailed Student's $t$-test. All error bars are expressed as mean values ± SD. Source data are provided as a Source Data file.

function in the absence of Cpt1a expression. Moreover, we showed that expression of human MPC2, resistant to shMpc2 targeting, partially restores proliferation in Cpt1a-deficient cells with stable Mpc2 silencing (Supplementary Fig. 4f). This effect was reversed by MPC inhibitor treatment (Supplementary Fig. 4g). In contrast, wild-type ErbB2+ cells, which are not reliant on Mpc2, exhibited no significant changes in proliferation upon re-expression of human MPC2 or MPC inhibition (Supplementary Fig. 4f, g). Collectively, these findings strongly support the specificity of Mpc2 knockdown effects and the functional relevance of MPC in supporting pyruvate-dependent cell survival and residual proliferation in Cpt1a-deficient cells.

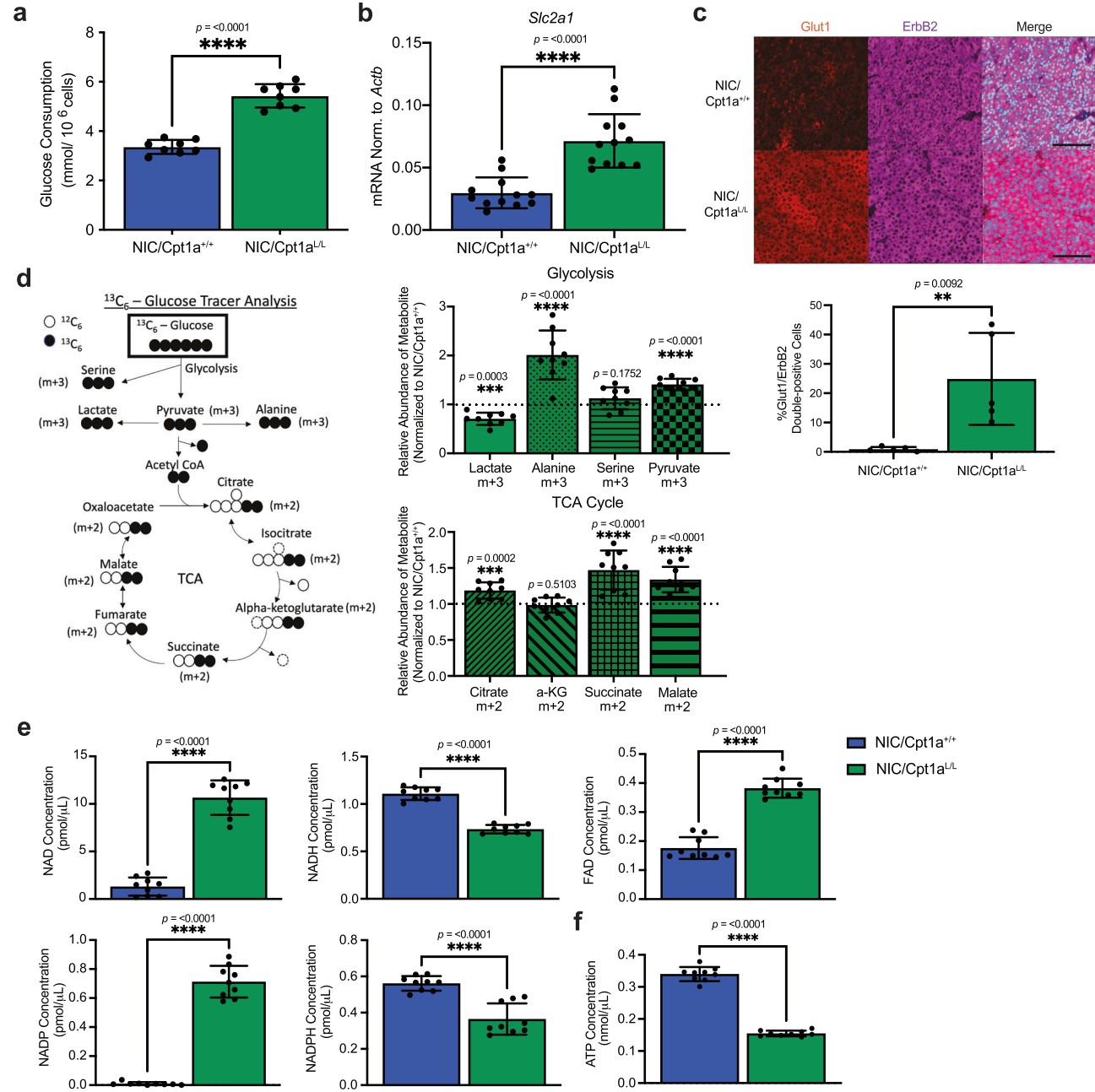

**Fig. 3 | Deletion of Cpt1a alters glucose metabolism and impairs the reduction of electron carriers. a** Glucose consumption in conditioned media from NIC/Cpt1a[+/+] and NIC/Cpt1a[L/L] cell lines (n = 4 cell lines per genotype in duplicate – ****$p < 0.0001$; unpaired, two-tailed Student's $t$-test). **b** Glut1 (Slc2a1) transcript levels in NIC/Cpt1a[+/+] and NIC/Cpt1a[L/L] cells normalized to Actb. n = 4 per genotype, analyzed in triplicate - ***$p < 0.001$; unpaired, two-tailed Student's $t$-test. **c** NIC tumors were immunostained with the indicated antibodies and DAPI. Right panel - Images representative of ten independent mice of each genotype. Scale bar represents 100 μm. Left panel - Staining was quantified and normalized to total cell number (DAPI). Minimum 10,000 cells per sample, **$p < 0.01$; unpaired, two-tailed Student's $t$-test. **d** Fractional ion abundance of glycolytic (Lactate, Alanine, Serine, and Pyruvate) and TCA Cycle (Citrate, a-KG, Succinate, Malate) intermediates following a pulse in U-[13]C-glucose of NIC/Cpt1a[L/L] cells normalized to wild-type (NIC/Cpt1a[+/+]) cells. n = 3 per genotype, analyzed in triplicate - ***$p < 0.001$, ****$p < 0.0001$; unpaired, two-tailed Student's $t$-test. **e** Relative abundance of oxidized (nicotinamide adenine dinucleotide - NAD[+], nicotinamide adenine dinucleotide phosphate - NADP[+] and flavin adenine dinucleotide - FAD) and reduced (NADH and NADPH) forms of electron carriers in NIC/Cpt1a[+/+] and NIC/Cpt1a[L/L] cells. n = 3 per genotype, analyzed in triplicate - ****$p < 0.0001$; unpaired, two-tailed Student's t-test. **f** ATP Concentration in NIC/Cpt1a[+/+] and NIC/Cpt1a[L/L] cells. n = 3 per genotype, analyzed in triplicate - ****$p < 0.0001$; unpaired, two-tailed Student's t-test. All error bars are expressed as mean values ± SD. Source data are provided as a Source Data file.

Despite the increased entry of pyruvate into the TCA cycle, OXPHOS was significantly suppressed in Cpt1a-null cells (Fig. 2c). This likely reflects the reduced entry of acetyl-CoA derived from FAO into the TCA cycle. However, FAO also directly generates NADH and FADH₂ from NAD[+] and FAD[+], providing a source of reducing equivalents required for OXPHOS[14]. We observed that Cpt1a-deficient tumor cells had elevated cellular ratios of NAD[+]/NADH and NADP[+]/NADPH, coupled with reduced ATP levels (Fig. 3e, f and Supplementary Fig. 5a). This may be attributed to decreased flux through the TCA cycle and the loss of NADH derived directly from FAO.

To further investigate the relative contribution of glucose-derived carbon (via pyruvate) and fatty-acid derived carbon (from acetyl-CoA) to overall TCA flux and subsequent OXPHOS and ATP generation in ErbB2+ cells with and without Cpt1a, we employed specific inhibitors— UK-5099, targeting MPC, and Etomoxir, which inhibits Cpt1a. Treatment of wild-type ErbB2+ cells with Etomoxir, significantly reduced levels of NADH, NADPH, ATP, and oxygen consumption, reflecting diminished OXPHOS, whereas UK-5099 treatment had no effect (Supplementary Fig. 5b–d). However, in Cpt1a-deficient cells, inhibition of MPC by UK-5099 severely reduced NADH and NADPH levels, resulting in decreased OCR and ATP generation. Notably, etomoxir had no significant effect on Cpt1a knockout cells under these conditions (Supplementary Fig. 5b–d). These findings highlight the crucial role of MPC in supporting TCA cycle flux, OXPHOS, and ATP production when FAO is impaired. Understanding these metabolic adaptations has implications for developing targeted therapies for cancer, where such metabolic vulnerabilities might be exploited for therapeutic benefit. These observations also argue that compromised reduction of $NAD^+$ contributes to the reduced oxygen consumption rate (OCR) and ATP production in Cpt1a-deficient cells compared to controls.

### Cpt1a ablation induces oxidative stress and associated responses in ErbB2+ breast cancer cells

An additional consequence of increased $NAD^+$/NADH and NADP+/ NADPH ratios is redox imbalance, which can lead to oxidative stress and associated oxidative damage to macromolecules including DNA, lipids, and proteins resulting in cell death[43,44]. We observed a pronounced increase in both the intensity and the percentage of cells labeled with a dye that detects reactive oxygen species (ROS), the cause of oxidative stress, in ErbB2+ cells lacking Cpt1a (Fig. 4a)[45]. Transcriptomic analysis (RNA-Seq) revealed that the 2614 transcripts up-regulated in NIC/Cpt1a$^{L/L}$ cells compared to NIC/Cpt1a$^{+/+}$ controls were dominated by oxidative stress response genes (Fig. 4b, Supplementary Fig. 6a). Further bioinformatic analysis of the transcriptomic data revealed that Nuclear factor erythroid 2-related factor 2 (Nfe2r2, also known as Nrf2), a critical regulator of cellular defence against toxicity and oxidative damage, was the transcription factor most significantly affected by Cpt1a-deficiency (Fig. 4b)[46]. Through unsupervised hierarchal clustering, we confirmed that an Nrf2 target gene signature, including many key regulators of the response to oxidative stress, metabolic genes involved in glucose consumption, including *Glut1*, as well as NADPH metabolism and the pentose phosphate pathway, is significantly upregulated in cells lacking Cpt1a (Fig. 4b, c)[47–49]. Cpt1a ablation led to elevated Nrf2 expression at the protein and mRNA levels in tumor cells in vitro and in vivo (Fig. 4d and Supplementary Fig. 6b), and we validated the elevated expression of Nrf2 regulated genes in Cpt1a-deficient NIC cells compared to controls (Fig. 4e). Collectively, these data argue that loss of Cpt1a disrupts redox homeostasis in ErbB2+ tumor cells, triggering an oxidative stress response through upregulation of Nrf2 and its target genes.

Furthermore, the mitochondria of Cpt1a-null ErbB2 tumor cells were also more numerous and of diminished length, whereas wild-type cells exhibited a highly fused mitochondrial network, consistent with the effects of ROS and oxidative stress on mitochondrial fragmentation (Supplementary Fig. 7a)[50]. Overall, these findings indicate that cells undergo metabolic reprogramming in the absence of Cpt1a, with an increased reliance on glucose to fuel the TCA cycle. However, the loss of FAO-derived acetyl-CoA and reducing equivalents strongly suppresses OXPHOS and triggers oxidative stress, increasing ROS production and causing mitochondrial fragmentation, which may further suppress mitochondrial metabolism and OXPHOS in Cpt1a-deficient ErbB2-driven tumor cells.

To assess whether we could mitigate oxidative stress and its downstream effects in cells without Cpt1a, we treated wild-type and Cpt1a-deficient ErbB2+ cells with N-acetylcysteine (NAC), a well-known antioxidant and ROS scavenger. Notably, the protective effects of NAC have been associated with the restoration of mitochondrial homeostasis through upregulation of mitochondrial fusion regulators such as Mfn1 and Opa1, and the suppression of mitochondrial fission, as indicated by decreased levels of the fission protein Drp1[51,52]. Strikingly, NAC treatment partially restored mitochondrial fusion in ErbB2+ cells lacking Cpt1a (Supplementary Fig. 7b). This was further associated with partial restoration of ATP levels, OXPHOS, and cell proliferation in Cpt1a-deficient cells in the presence of NAC (Supplementary Fig. 7c-e). Together these data indicate that NAC treatment can alleviate oxidative stress and reinstate redox balance within the mitochondria. Moreover, while some Nrf2 target gene are downregulated upon NAC treatment, NAC can promote the upregulation of *Nrf2* itself, subsequently increasing *Glut1* expression and glucose uptake in cells lacking Cpt1a (Supplementary Fig. 7f–h)[53]. Additionally, NAC promotes glucose carbon utilization by the TCA cycle, favoring mitochondrial function over lactate synthesis (Supplementary Fig. 7i). These results support the conclusion that Cpt1a-deficient cells undergo excessive oxidative stress and rely on Nrf2 and glucose oxidation for OXPHOS and energy production, and their proliferation can be rescued by antioxidant treatment.

### Upregulation of Nrf2 is an adaptive mechanism in the absence of Cpt1a

To evaluate the importance of Nrf2 in Cpt1a-deficient cells, we employed two potent inhibitors of Nrf2, Brusatol and ML385, with distinct mechanisms of action[54,55]. Brusatol provokes a rapid and transient depletion of Nrf2 protein, while ML385 interacts with Nrf2 and affects its DNA binding activity[54,55]. Nrf2 inhibition in combination with Cpt1a ablation significantly reduced tumor cell proliferation (Fig. 5a and Supplementary Fig. 8a). In contrast to wild-type counterparts, which exhibited decreased dependence on Nrf2 for glucose uptake, inhibition of Nrf2 further diminished glucose consumption and *Glut1* expression in Cpt1a-deficient cells (Fig. 5b, c and Supplementary Fig. 8b). The suppression of mitochondrial bioenergetics in Cpt1a-deficient cells was exacerbated by inhibition of Nrf2, accompanied further by reduced basal OCR and ECAR, whereas Nrf2 inhibitors exerted minimal effects on wild-type cells (Fig. 5d, e).

Consistent with the inhibitor studies, stable silencing of Nrf2 suppressed proliferation, glucose consumption, and the basal and maximal OCRs in Cpt1a-deficient ErbB2+ tumor cells (Fig. 5f–j, and Supplementary Fig. 8c, e), but had minimal effects on wild-type cells, arguing that Nrf2 is present but not essential in these cells. Conversely, constitutive activation of Nrf2 by knockdown of its repressor, Keap1[56], promoted cell growth, glucose uptake and mitochondrial OXPHOS in cells lacking Cpt1a (Fig. 5f–j and Supplementary Fig. 8d, e). This partially restored OCR, ECAR, glucose consumption, and in Cpt1a-deficient cells, while effects on wild-type cells were of a lesser magnitude.

Collectively, these results indicate that Cpt1a-deficient cells rely on Nrf2 to promote glucose consumption, revealing a metabolic vulnerability that can be exploited by pharmacological or genetic targeting of these pathways in breast cancers.

### Targeting Cpt1a sensitizes ErbB2-driven breast tumors to the effects of a ketogenic diet

Although the synthetic lethal combination of Cpt1a and Nrf2 inhibition seems appealing, the translational potential of this approach may be limited by the absence of clinical standardization and the potential systemic toxicity associated with Nrf2 inhibitors[57]. An alternative approach to target increased glucose dependency, that does not rely on direct inhibition of Nrf2, is therefore warranted.

Tumor cell growth media used for in vitro studies contains supraphysiological levels (25 mM) of glucose. To examine glucose

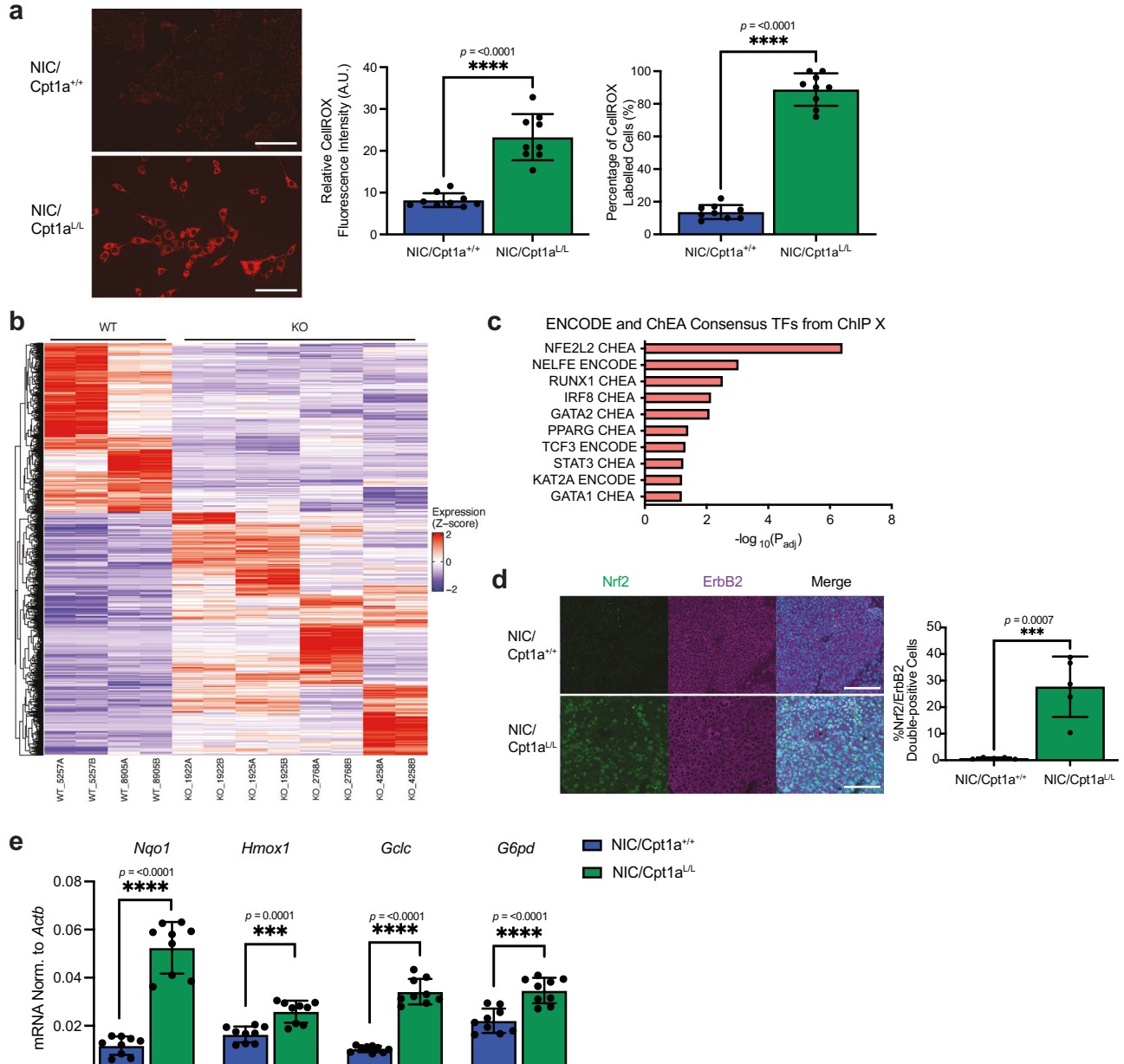

**Fig. 4 | Cpt1a-deficient cells undergo oxidative stress and upregulate Nrf2 expression and activity in vivo. a** Left panel - Representative fluorescence microscope image of NIC/Cpt1a$^{+/+}$ and NIC/Cpt1a$^{L/L}$ cells with CellROX Orange reagent. Scale bar represents 200 μm. Right panel – Quantification of relative CellROX Fluorescence Intensity (absorgance units - A.U.) and percentage of Cell-ROX labeled cells. $n = 3$ cell lines per genotype ***$p < 0.001$; unpaired, two-tailed Student's $t$-test. **b** Unsupervised hierarchical clustering analysis of genes differentially up-regulated (red) and down-regulated (blue) in NIC/Cpt1a$^{L/L}$ cells compared to NIC/Cpt1a$^{+/+}$ controls ($n = 2$ or 4 per genotype, analyzed in duplicate). **c** ENCORE and ChEA Consensus Transcription Factor analysis of up-regulated transcription factors in Cpt1a$^{L/L}$ cells compared to wild-type NIC cells using Enrichr. **d** Left panel –

End-stage tumors were stained with Nrf2 and ErbB2-specific antibodies and DAPI (nuclei). Scale bar represents 100 μm. Right panel – Nrf2 immunostaining was quantified and normalized to ErbB2-positive cells. $n = 5$ mice per genotype (minimum 10,000 total nuclei analyzed per mouse). ***$p < 0.001$; unpaired, two-tailed Student's $t$-test. **e** Quantitative real-time polymerase chain reaction (QRT-PCR) analysis of Nrf2 target gene expression in NIC/Cpt1a$^{+/+}$ and NIC/Cpt1a$^{L/L}$ cells. Nrf2 target expression was normalized to that of *Actb*. $n = 3$ cell lines per genotype in triplicate - ***$p < 0.001$, ****$p < 0.0001$; unpaired, two-tailed Student's $t$-test. All error bars are expressed as mean values ± SD. Source data are provided as a Source Data file.

dependency in ErbB2+ tumor cells further, we performed glucose-limiting growth assays including a range of conditions mimicking extracellular conditions in normal mammary tissue (2.5 mM) and in poorly vascularized tissue (1 mM)[58]. Restricting glucose availability reduced the proliferation of both Cpt1a-proficient and -deficient ErbB2+ breast tumor cells in a concentration-dependent manner. In contrast to Cpt1a-proficient cells, low glucose conditions completely suppressed the growth of tumor cells lacking Cpt1a (Supplementary

Fig. 9a). Supplementation with the long-chain fatty acid, palmitic acid, or the ketone, 3-hydroxybutyrate (3-OHB), partially rescued cell growth in wild-type ErbB2+ cells under low glucose conditions (Fig. 6a and Supplementary Fig. 9b, c) and promoted growth under high-glucose conditions. In contrast, Cpt1a-deficient cells did not respond to palmitate or ketone body supplementation, consistent with our other findings and suggesting a loss of the metabolic flexibility that enables lipid utilization for energy production under low-glucose

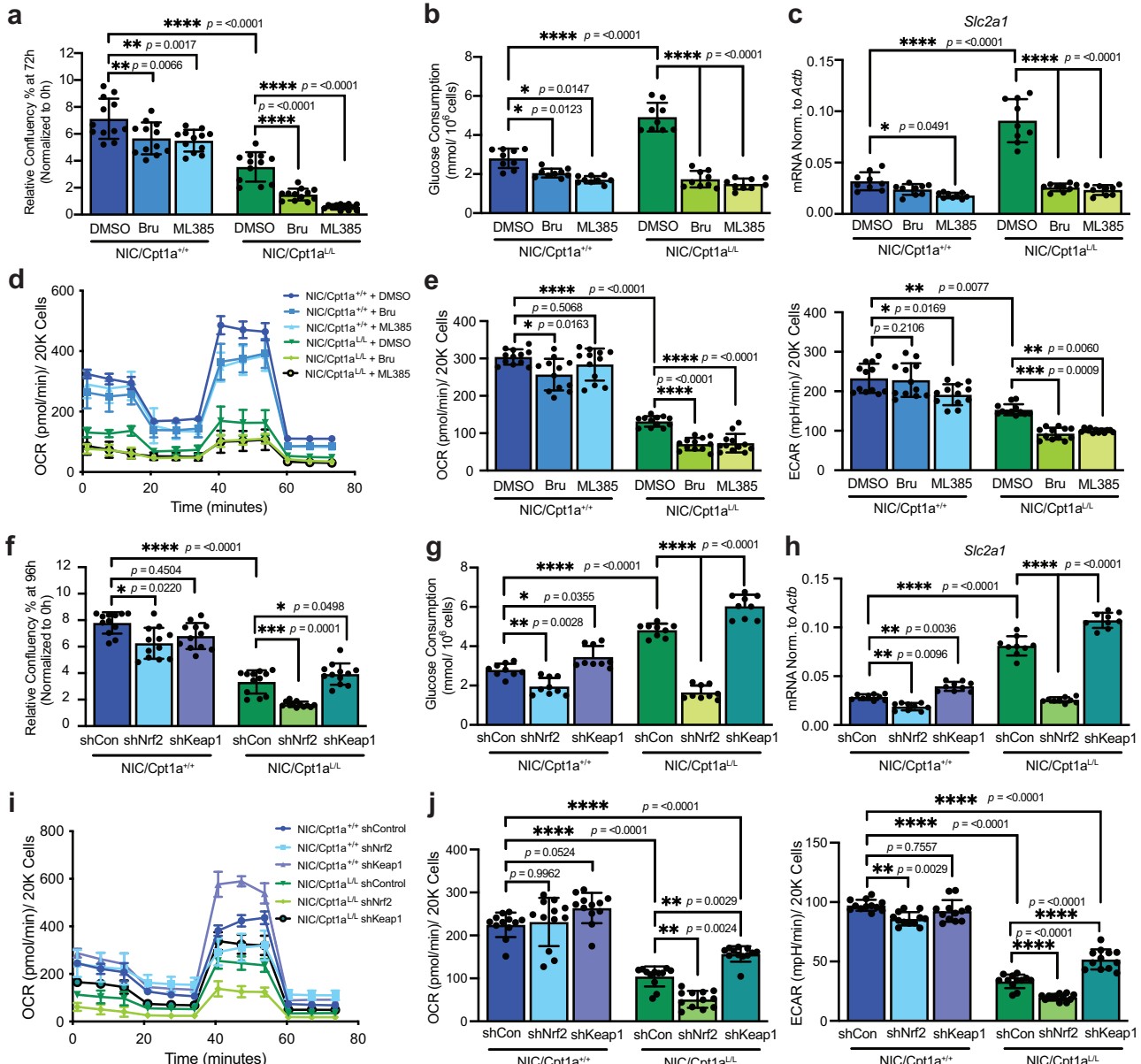

**Fig. 5 | Nrf2 modulates glucose uptake and metabolism in Cpt1a-deficient cells.** **a** NIC cells were treated with the Nrf2 inhibitors, Brusatol (100 nM), ML385 (2µM)), or vehicle (DMSO). Proliferation was assessed by Incucyte after 72 h. Data normalized to confluency at $t = 0$. $n = 3$ cell lines in triplicate · **$p < 0.01$, ****$p < 0.0001$; one-way ANOVA with Tukey's post-hoc test. Glucose consumption (**b**) and QRT-PCR analysis of *Glut1* gene expression (**c**) in cells treated as in (**a**). $n = 3$ cell lines per genotype in triplicate – *$p < 0.05$, ****$p < 0.0001$; one-way ANOVA with Tukey's post-hoc test. **d** Basal, maximal (Carbonyl cyanide 4-(trifluoromethoxy)phenylhydrazone – FCCP), ATP-synthesis coupled (Oligomycin A), and non-mitochondrial (rotenone/ antimycin A) oxygen consumption rates (OCRs) of NIC cells treated with DMSO or Nrf2 inhibitors. Representative of $n = 3$ cell lines per genotype per condition. **e** Quantification of basal OCR and extracellular acidification rates (ECAR) of cells as in (**a–e**). $n = 3$ cell lines per genotype in quadruplicate – *$p < 0.05$, **$p < 0.01$,

***$p < 0.001$, ****$p < 0.0001$; one-way ANOVA with Tukey's post-hoc test. **f** NIC cells stably expressing short hairpin RNAs (shRNAs) against luciferase (control – shCon), Nrf2 or Keap1. Proliferation assessed using Incucyte after 96 h. $n = 3$ cell lines per genotype in triplicate · *$p < 0.05$, ***$p < 0.001$, ****$p < 0.0001$; one-way ANOVA with Tukey's post-hoc test. Glucose consumption (**g**) and QRT-PCR analysis of *Glut1* expression (**h**) in cells as in (**f**) transduced with indicated shRNAs. $n = 3$ cell lines per genotype in triplicate – *$p < 0.05$, **$p < 0.01$, ****$p < 0.0001$; one-way ANOVA with Tukey's post-hoc test. **i** OCR analysis of NIC cell lines expressing control, Nrf2 or Keap1 shRNAs as in (**d**). Representative of $n = 3$ cell lines per genotype. **j** Quantification of basal OCR and ECAR of cells as in (**e, f**). $n = 3$ cell lines per genotype in triplicate –**$p < 0.01$, ****$p < 0.0001$; one-way ANOVA with Tukey's post-hoc test. All error bars are expressed as mean values ± SD. Source data are provided as a Source Data file.

conditions (Fig. 6a and Supplementary Fig. 9b, c). At the highest concentrations of palmitic acid and 3-OHB tested, proliferation of cells of both genotypes was suppressed, likely due to toxicity as previously described in other cell types[29,59]. In addition, although 3-OHB can enter the mitochondria independently of Cpt1a, the metabolic state of the cell may affect its utilization. Specifically, the expression of *Bdh1*, responsible for metabolizing 3-OHB, is significantly lower in Cpt1a-deficient cells (Supplementary Fig. 9d). Moreover, reduced Cpt1a

levels limit acetyl CoA availability, a precursor for ketone body synthesis, further decreasing Bdh1 activity (Supplementary Fig. 9e). Consequently, 3-OHB supplementation and ketone oxidation do not compensate for Cpt1a deficiency in ErbB2+ cells.

We next investigated the potential efficacy of combining FAO inhibition with a ketogenic diet, rich in long-chain fatty acids, using an orthotopic allograft model in mice (Fig. 6b). Our rationale was to decrease glucose availability and eliminate the possibility of using FAO

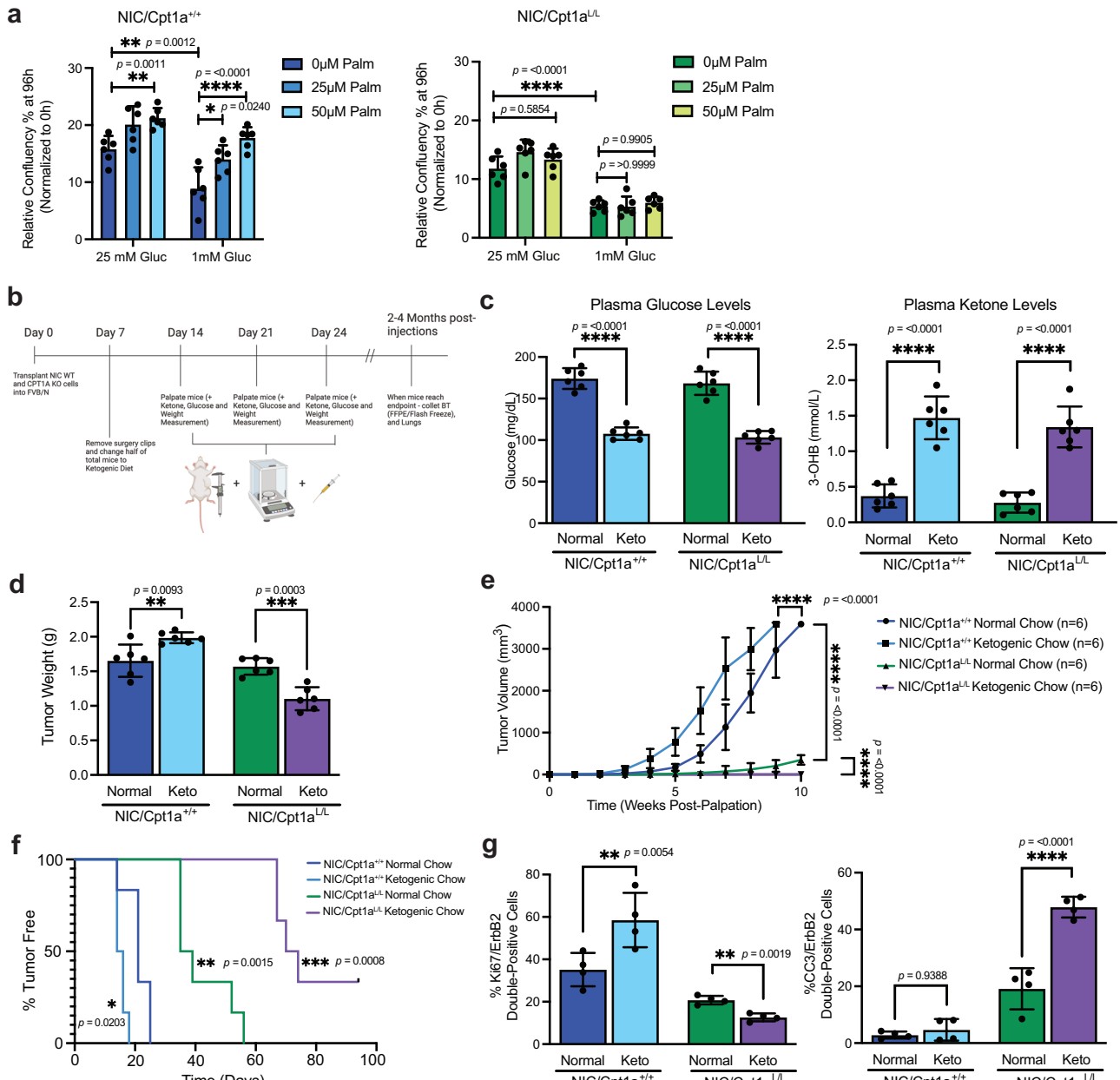

**Fig. 6 | Proliferation of Cpt1a-deficient cells is compromised in high-lipid, low-glucose conditions in vitro or in vivo. a** Proliferation was assayed in NIC cells supplemented with palmitate under variable glucose concentrations for 96 h. Data were normalized to confluency at t = 0. $n = 3$ cell lines per genotype in triplicate – *$p < 0.05$, **$p < 0.01$, ****$p < 0.0001$; one-way ANOVA with Tukey's post-hoc test. **b** Schematic showing the experimental design of the ketogenic diet orthotopic xenotransplant study using two independent NIC/Cpt1a$^{+/+}$ and NIC/Cpt1a$^{L/L}$ cell lines. Created with BioRender.com **c** Plasma glucose and ketone levels were measured. $n = 6$ per group, ****$p < 0.0001$; one-way ANOVA with Tukey's post-hoc test. **d** Tumor mass was assessed at endpoint. $n = 6$ mice per treatment group, **$p < 0.01$, ***$p < 0.001$; one-way ANOVA with Tukey's post-hoc test. **e** Tumor burden was determined by weekly caliper measurements till control mice (wild-type NIC tumors on normal chow) reached endpoint (10 weeks). $n = 6$ mice per treatment group, ****$p < 0.0001$; one-way ANOVA with Tukey's post-hoc test. **f** Kaplan–Meier survival analysis. $n = 6$ mice per treatment group, *$p < 0.05$, **$p < 0.01$, ***$p < 0.001$; log rank test. **g** Quantification of Ki67, cleaved caspase 3 and ErbB2 immunostaining in endpoint tumors. **$p < 0.01$, ****$p < 0.0001$; one-way ANOVA with Tukey's post hoc-test. All error bars are expressed as mean values ± SD. Source data are provided as a Source Data file.

as an alternative energy source, thereby rendering ErbB2+ cells more susceptible to cell death through nutrient starvation (Supplementary Fig. 10a). Mice harboring NIC/Cpt1a$^{+/+}$ and NIC/Cpt1a$^{L/L}$ tumors exhibited decreased total body weight, reduced plasma glucose, and increased 3-OHB levels on the ketogenic diet compared to control diet, confirming that ketosis was achieved in these animals (Fig. 6c and Supplementary Fig. 10b). Interestingly, the ketogenic diet increased tumor size and decreased survival in mice bearing wild-type ErbB2+ tumors, consistent with the efficient use of lipids and ketone bodies as

metabolic fuels for NIC/Cpt1a$^{+/+}$ tumor cells in vitro and in vivo (Fig. 6d–f, and Supplementary Fig. 10c). In contrast, combining Cpt1a ablation with the ketogenic diet markedly decreased tumor growth and enhanced survival compared to all other conditions. Moreover, Cpt1a-deficient tumors on the ketogenic diet displayed reduced proliferation (Ki67) and increased apoptosis (cleaved caspase-3) compared to normal diet controls and Cpt1a-proficient ErbB2+ tumors treated with either a standard diet or a ketogenic diet (Fig. 6g and Supplementary Fig. 10d). Lung metastases were also attenuated in

Cpt1a-deficient, ketogenic diet–treated cohorts (Supplementary Fig. 10e). In addition to the effect of the ketogenic diet on fatty acid transporters, we observed an enhanced expression of *Cd36* in both Cpt1a-proficient and -deficient groups, with a more pronounced effect observed in the control, wild-type tumors (Supplementary Fig. 10f). Additionally, the ketogenic diet significantly diminished glucose uptake through downregulating the expression of *Nrf2* and the glucose transporter, *Glut1*, in Cpt1a-deficient tumors (Supplementary Fig. 10f). In contrast, mice transplanted with wild-type ErbB2+ cells, which exhibit metabolic plasticity allowing them to use multiple fuel sources, did not exhibit changes in *Nrf2* and *Glut1* expression. Collectively, these observations argue that, while ketosis can exacerbate ErbB2+ tumor growth, combining it with FAO blockade via Cpt1a targeting is an effective therapeutic strategy. Nevertheless, caution is warranted when considering dietary therapeutic approaches, particularly in scenarios where the ketogenic diet alone may promote ErbB2+ breast tumor growth.

Given the role of Cpt1a in regulating long-chain fatty acid (LCFA) mitochondrial entry and oxidation, it is conceivable that medium-chain fatty acids (MCFAs) could circumvent the Cpt1a blockade and rescue proliferation defects observed upon Cpt1a ablation in HER2+ cells. To investigate this, we supplemented Cpt1a-null cells with various MCFAs, including Caprylic acid (C8:0), Capric acid (C10:0), and Lauric acid (C12:0), all of which rescued proliferation in Cpt1a-deficient cells (Supplementary Fig. 11a). Supplementation with lauric acid, abundant in coconut oil – a prevalent source of medium-chain triglycerides – partially restored migration and invasion, as well as energy production through OXPHOS, in Cpt1a-deficient cells (Supplementary Fig. 11b, c). However, even with lauric acid supplementation, these cells exhibited partial reliance on Nrf2 and glucose consumption, albeit at lower levels than Cpt1a-deficient cells without addition of MCFAs (Supplementary Fig. 11d, e). Our findings argue that Cpt1a-deficient cells can utilize both glucose and MCFAs as alternative fuel sources, highlighting their metabolic flexibility in adapting to changes in nutrient availability.

To evaluate whether these findings could be translated to an in vivo model, we formulated a Medium-Chain Ketogenic Diet (MCKD) where LCFAs were replaced by MCFAs, primarily sourced from coconut oil, while the remaining components were identical to the Long-Chain Ketogenic Diet (LCKD) (Supplementary 10a and Supplementary Table S2). In mice bearing NIC/Cpt1a$^{+/+}$ and NIC/Cpt1a$^{L/L}$ tumors, the MCKD effectively induced ketosis, evidenced by reduced plasma glucose, increased 3-OHB levels and decreased total body weight compared to the control diet (Supplementary Fig. 11f, g). As observed with the LCKD, the MCKD increased tumor volume and decreased survival in wild-type tumors. However, whereas the combination of Cpt1a ablation and LCKD significantly inhibited tumor growth and enhanced survival (Fig. 6d–f), Cpt1a-deficient tumors on the MCKD grew at a comparable rate to those on normal chow (Supplementary Fig. 11h, i), consistent with the ability of MCFAs to partially rescue the effects of glucose starvation on proliferation and metabolism in vitro. Nonetheless, the MCKD was not able to rescue tumor growth in vivo to the level of Cpt1a-proficient ErbB2+ tumor cells (Supplementary Fig. 11h, i). These results argue that Cpt1a-deficient cells can adapt to their inability to metabolize LCFAs by using MCFAs to support cell survival and growth under low glucose/ketogenic diet conditions. However, they remain severely impaired in comparison to Cpt1a-proficient cells under these conditions. Overall, our data are consistent with the therapeutic potential of a ketogenic diet abundant in LCFAs combined with FAO blockade in ErbB2+ breast cancer.

## ErbB2-targeted therapy response is enhanced by Cpt1a deletion in an immunocompetent ErbB2+ breast cancer GEMM

Using RNA expression data from both microarray and mRNA sequencing, we confirmed that overexpression of *CPT1A* is a marker of poor prognosis correlating with decreased overall survival in all breast cancer patients ($p = 0.0035$) and specifically in HER2+ breast tumors ($p = 0.00071$) (Fig. 7a). We also investigated whether CPT1A expression was implicated as a biomarker of resistance to HER2-targeted therapy, particularly Trastuzumab, in HER2+ breast cancer. Strikingly, Trastuzumab non-responsive tumors exhibit significantly elevated levels of CPT1A compared to the baseline and responsive tumors, potentially implicating CPT1A activity in Trastuzumab resistance (Fig. 7b).

Elevated CPT1A activity has been implicated in promoting metabolic flexibility that leads to ErbB2-targeted therapy resistance in breast cancer[18,25]. Increased Cpt1a activity enables cancer cells to utilize FAs as an energy source, bypassing the reliance on ErbB2 signaling for growth and survival[25]. Consequently, these metabolic adaptations allow the cancer cells to evade the inhibitory effects of ErbB2-targeted therapies, which include suppression of glucose metabolism and energetic stress. To test the role of lipid metabolism in targeted therapy responses in our model, we first treated wild-type ErbB2+ cells with lapatinib under varying glucose conditions and FA or ketone supplementation. We observed the strongest growth inhibition under low glucose conditions, which could be rescued by supplementation with either FAs or ketone bodies (Fig. 7c). This finding was consistent with the potential of targeting Cpt1a or modulating FA metabolism alongside HER2-targeted therapy to enhance treatment efficacy in therapy-resistant HER2-positive breast cancer.

To investigate this further in the context of ErbB2 mAb therapy, we used an immunocompetent, orthotopic ErbB2+ allograft model and the anti-ErbB2 mAb clone, 7.16.4, which recognizes an epitope of rodent ErbB2 that overlaps with the Trastuzumab-binding epitope of HER2[60]. In pre-clinical models employing two independently derived cell lines, proficient and deficient in Cpt1a, we observed that combining 7.16.4 mAb treatment and Cpt1a deletion significantly attenuated tumor growth and proliferation and elevated tumor cell apoptosis, compared to all other conditions (Fig. 7d–f, and Supplementary Fig. 12a). Additionally, the combined 7.16.4 mAb treatment and Cpt1a deletion reduced the size and number of lung metastatic lesions, indicating the potential efficacy of this combination therapy in both primary and metastatic disease (Supplementary Fig. 12b). Supporting the involvement of metabolic regulators in these observations, 7.16.4 mAb treatment decreased the expression of *Nrf2* and the glucose transporter, *Glut1*, whilst upregulating *Cd36* expression (Supplementary Fig. 12c). However, this was effect was reversed by Cpt1a deletion, which resulted in decreased expression of both transporters.

As the effects of anti-ErbB2 mAb therapy are also mediated by the immune system, through mechanisms including antibody-dependent cellular cytotoxicity (ADCC), we investigated whether modulating FA metabolism affected anti-tumor immune responses in our model[61–63]. Interestingly, combined 7.16.4 mAb and Cpt1a inhibition elicited the highest levels of STAT1 phosphorylation (p-Stat1 (Y701)), suggesting the strongest induction of an interferon-driven immune response (Fig. 7g)[64]. Additionally, the infiltration of M1-polarized tumor-associated macrophages (TAMs - F4/80$^+$, p-Stat1 (Y701)$^+$) was increased in the combination therapy group, which is often correlated with improved survival and outcome in breast cancer patients (Fig. 7g)[65–67]. These data are consistent with an enhanced anti-tumor immune response to Cpt1a ablation and 7.16.4 mAb combination therapy compared to other conditions tested. We confirmed that, while the recruitment of CD4$^+$ T cells was unaffected, tumor-infiltrating CD8$^+$ cytotoxic T cells and natural killer (NK) cells was significantly enriched by Cpt1a inhibition and 7.16.4 mAb treatment (Supplementary Fig. 12d). Furthermore, tumor infiltration by IFNγ-expressing CD8 + T cells was increased in the Cpt1a ablated/7.16.4 mAb-treated tumors compared with control groups, consistent with the recruitment and activation of cytotoxic, anti-tumor leukocytes (Supplementary Fig. 12e, f). Collectively, these findings suggest that metabolic reprogramming through targeting Cpt1a promotes an antitumor immune

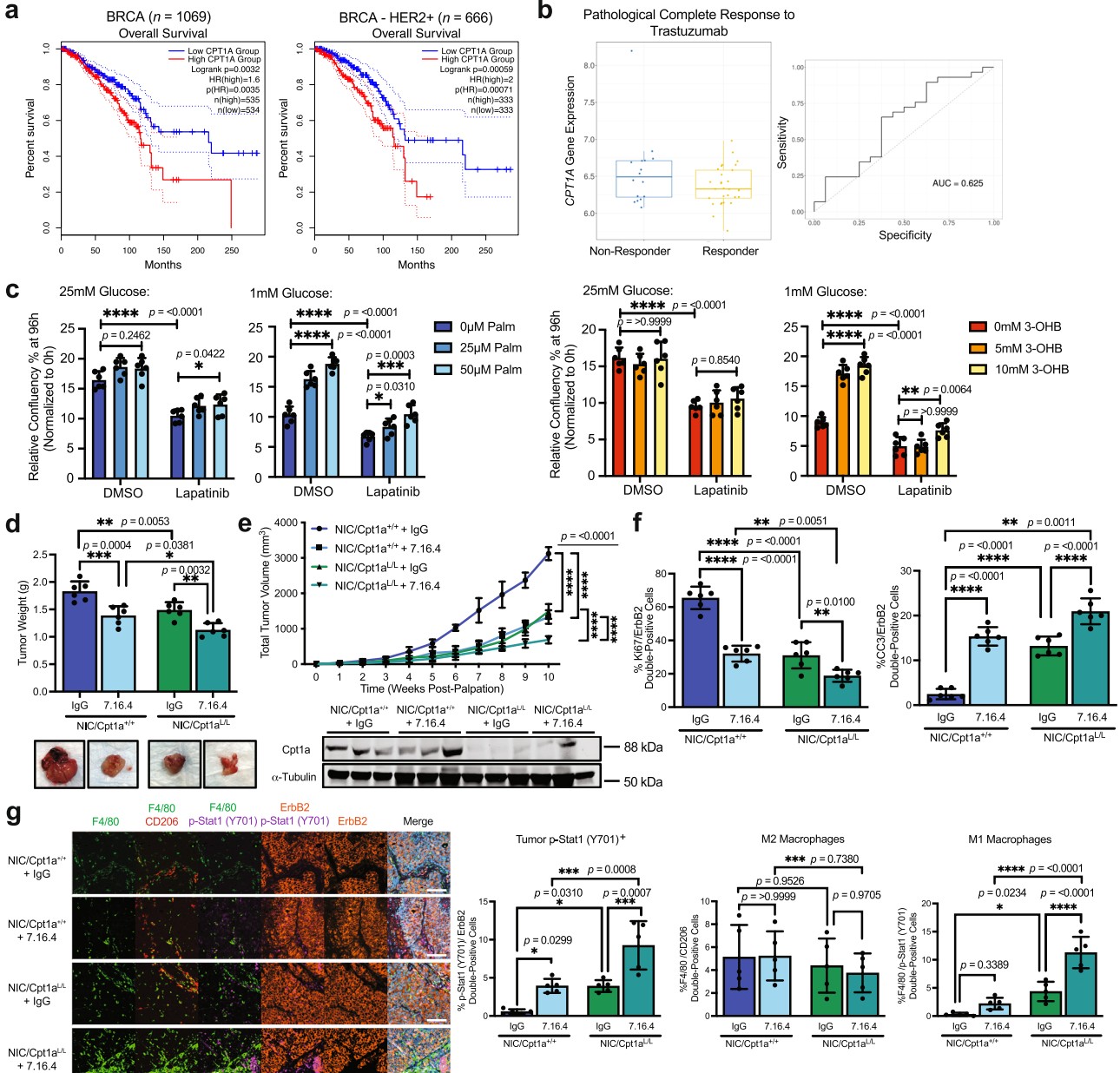

**Fig. 7 | Cpt1a loss enhances ErbB2 monoclonal antibody response in tumor suppression. a** Kaplan–Meier survival curve of Breast Cancer (BRCA) ($n = 1069$) and HER2-positive BRCA ($n = 666$) patients with high or low *CPT1A* expression (log rank $p = 0.0032$ and $p = 0.00059$, respectively) segregated using median cut-off. **b** ROC-plotter analysis of *CPT1A* gene expression in Trastuzumab non-responder and responders ($n = 120$, ROC *p*-value = 1.2 e-02). Box plot with center line = median, box = 25th–75th quartile, whiskers = maxima/minima, outliers = open circle. **c** Proliferation assay in NIC cells treated with DMSO or Lapatinib supplemented with Palmitate (left panel) or Ketone (3-OHB, right panel) at varying glucose concentrations. Data normalized to confluency at $t = 0$. $n = 2$ cell lines per genotype in triplicate – *$p < 0.05$, **$p < 0.01$, ***$p < 0.001$, ****$p < 0.0001$; one-way ANOVA with Tukey's post-hoc test. **d** Immunocompetent mice bearing orthotopic NIC/Cpt1a[+/+] and NIC/Cpt1a[L/L] breast cancer allografts treated with IgG (control Ab) and 7.16.4 Ab ($n = 10$ per treatment group). Tumor weight (top panel) and representative images

(bottom panel) at end-point. $n = 10$ mice per treatment group, *$p < 0.05$, **$p < 0.01$ and ***$p < 0.001$; one-way ANOVA with Tukey's post-hoc test. **e** Top panel – Weekly tumor volume assessment during treatment. $n = 10$ mice per treatment group, **$p < 0.01$,***$p < 0.001$ and ****$p < 0.0001$; one-way ANOVA with Tukey's post-hoc test. Bottom panel – immunoblot analysis of Cpt1a levels in endpoint tumors. **f** Quantification of Ki67 and cleaved caspase 3 by immunostaining endpoint tumors. **$p < 0.01$, ***$p < 0.001$, ****$p < 0.0001$; one-way ANOVA with Tukey's post hoc-test. **g** Left panel - End-stage NIC tumors treated with IgG and 7.16.4 mAb immunostained with immune cell markers and DAPI. Images representative of 10 mice per treatment group. Scale bar: 100 μm. Right panel - Quantification of p-Stat1(Y701)[+]/ErbB2[+], F4/80[+]/CD206[+] and F4/80[+]/p-Stat1(Y701)[+] by HALO Analysis. *$p < 0.05$, ***$p < 0.001$, ****$p < 0.0001$; one-way ANOVA with Tukey's post hoc-test. All error bars are expressed as mean values ± SD. Source data are provided as a Source Data file.

microenvironment that potentiates the response to anti-ErbB2 mAb therapy.

Taken together, our findings suggest that Cpt1a inhibition can significantly inhibit tumor growth, promote apoptosis, and reduce the occurrence of lung metastases when combined with a ketogenic diet

or HER2 monoclonal antibody therapy. Loss of FAO induces oxidative stress, triggering upregulation of *Nrf2* and subsequently *Glut1* expression as an adaptive response allowing the cells to meet the energy demands of cell survival and proliferation through enhanced glucose uptake. This regulation of glucose and lipid transporters

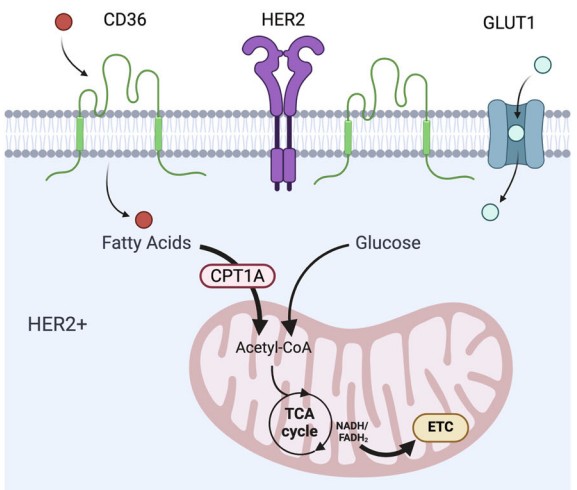
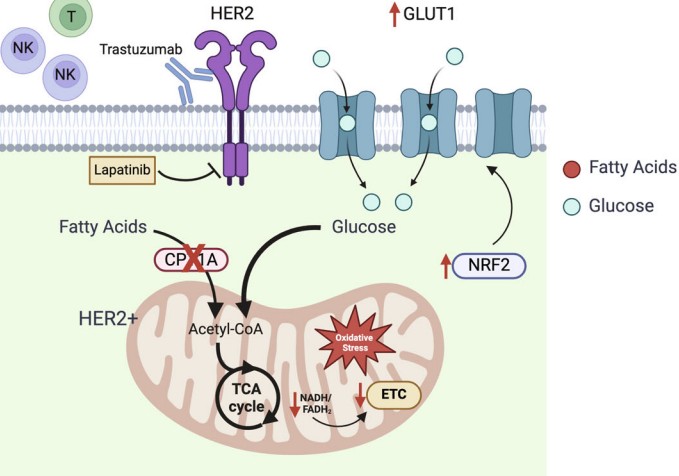

**Fig. 8 | Inhibition of Cpt1a sensitizes ErbB2-driven tumor cells to HER2-targeted therapy.** Schematic illustrating the role of lipid and glucose metabolism in energy production and proliferation in HER2+ cells. Disruption of fatty acid oxidation (FAO), mediated by Cpt1a, diminishes the reduction of electron carriers, impairs oxidative phosphorylation (OXPHOS) and promotes oxidative stress. Cpt1a-deficient cells exhibit increased glucose uptake and reprogram glucose metabolism to reflect a reliance on the TCA cycle for NAD + /FAD reduction. Inhibiting Cpt1a can enhance the response to the ketogenic diet or HER2 monoclonal antibody therapy by promoting an antitumor immune microenvironment, thereby improving the clinical outcome of HER2-positive breast cancer patients. Created with BioRender.com.

contributes to the enhanced sensitivity of tumor cells to the ketogenic diet and anti-ErbB2 mAb therapy in the absence of Cpt1a. These results offer valuable insights into the therapeutic efficacy of this combination approach for HER2-positive breast cancer patients and for those who develop resistance to standard therapy, both in primary and metastatic settings (Fig. 8).

## Discussion

Metabolic reprogramming has emerged as a pivotal aspect of cancer biology, with various metabolic enzymes explored as potential targets for cancer therapy. Despite substantial progress, our understanding of the vulnerabilities of specific tumor types to inhibitors, either as single agents or in combination with chemotherapy, radiation, dietary interventions and/ or immunotherapy remains incomplete. While glucose metabolism has traditionally been a major focus in cancer therapies, recent research has underscored the significance of FA metabolism in sustaining cancer cell proliferation and survival[68,69]. Consistent with these studies, we investigated the impact of mammary-epithelial deletion of Cpt1a in an ErbB2 GEMM, revealing a significant delay in tumor onset and reduced metastatic capacity (Fig. 1 and Supplementary Fig. 1). Metabolic and genetic analyses of Cpt1a-deficient tumors unveiled metabolic rewiring, with increased glucose import and shunting of glucose-derived pyruvate into the mitochondria to fuel the TCA cycle (Fig. 3a–d, Supplementary Figs. 3–5) associated with elevated Glut1, an insulin-independent glucose transporter, in Cpt1a-deficient cells (Fig. 3a–c).

Cpt1a-deficient ErbB2+ tumors also generated high levels of ROS, correlated with increased expression of Nrf2 (Fig. 4, Supplementary Fig. 6) and the induction of an Nrf2-dependant oxidative stress response (Fig. 4c–e). Notably, pharmacological, or genetic targeting of Nrf2 selectively impeded the proliferation of Cpt1a deficient ErbB2 + cells (Fig. 5a, f). Nrf2 is also a metabolic regulator with roles in key pathways including NADPH metabolism, the pentose phosphate pathway (PPP), and nucleotide synthesis, all of which have been implicated in promoting tumorigenesis[70]. Here, we demonstrate strong regulation of Glut1 expression by Nrf2, with Glut1 levels significantly reduced by genetic or pharmacological targeting of Nrf2 and elevated upon Nrf2 activation by antioxidant treatment or Keap1 knockdown (Fig. 5b, c, g, h, Supplementary Fig. 8b-d). These findings highlight the complex interplay between Nrf2-dependent cellular detoxification, lipid and glucose metabolism, unveiling potential avenues for therapeutic exploration.

While no change in Glut4 expression, which is insulin-dependent (Supplementary Fig. 3a), was observed, insulin signaling has been implicated in the translocation of both Glut1 and Glut4 to the cell membrane to facilitate glucose transport[71,72]. Elevated glucose uptake following Nrf2 activation raises interesting questions about the connection between insulin signaling and Nrf2, as well as the modulation of glucose transporters by Nrf2 activation. Insulin signaling has been reported to activate Nrf2, and both insulin and Nrf2 activation trigger cellular glucose uptake[73]. The causal connection and interdependency between these signaling pathways remain unresolved. However, it is possible that insulin, beyond its anabolic effects, may activate Nrf2, linking the cellular antioxidant response and detoxification with elevated glucose flux and increased metabolic activity. Targeting this signaling cascade in cancer cells may impair glucose utilization and adaptation to metabolic stress, especially in the context of underlying health conditions with chronically high blood glucose levels, such as diabetes, obesity, and insulin resistance. Further exploration of other metabolic consequences of Nrf2 activation and inhibition, such as effects on nucleotide synthesis, is also essential for understanding cancer cell adaptation to metabolic stress, including the response to targeting Cpt1a.

The ketogenic diet, characterized by high-fat, low-carbohydrate composition, has garnered attention as an adjuvant therapy in both preclinical and clinical trials in breast cancer[28,74,75]. Our study investigates the effects of the two ketogenic diets, one abundant in LCFAs (LCKD) and the other in MCFAs (MCKD), on breast cancer progression. We demonstrate that both the LCKD and the MCKD increased FA utilization, critical for ErbB2-positive breast cancer progression, and accelerated the progression of wild-type ErbB2+ tumors (Fig. 6g, Supplementary Fig. 10d, f, and Supplementary Fig. 11h, i). However, while the LCKD severely compromised the progression of tumors lacking Cpt1a and significantly increased animal survival in pre-clinical models (Fig. 6d–f), the progression of Cpt1a-deficient tumors on the

MCKD was largely indistinguishable from the normal rodent diet (Supplementary Fig. 11h, i). Nonetheless, Cpt1a-deficient tumors remained significantly impaired compared to wild-type controls under all conditions. These findings highlight the therapeutic potential of combining FAO inhibition in tumor cells with the ketogenic diet to prevent tumor progression and improve patient outcomes. However, they also emphasize the importance of considering the lipid composition of these diets, as well as encouraging caution around their use in the absence of strategies to block metabolic compensation (such as Cpt1a inhibition), which can enable the use of lipids as metabolic fuels to promote tumor growth[76–78].

The tumor microenvironment is a complex and dynamic milieu that is crucial in cancer progression and therapy response. Our study uncovered intriguing differences in the behavior of Cpt1a-deficient cells under in vitro and in vivo conditions. While these cells exhibited severely impaired proliferation in vitro, they demonstrated somewhat more potential for growth in vivo in genetically engineered models (Figs. 1c, d and 2a, b). This difference in behavior is potentially facilitated by metabolic support and cross-talk from the surrounding tumor microenvironment and highlights a dynamic interplay between cancer cells and the surrounding stromal cells, including adipocytes and fibroblasts, which may compensate for the nutrient deficiency and energy insufficiency upon the loss of fatty acid oxidation[79,80]. Future studies aiming to elucidate these interactions are clearly warranted to unveil the intricate metabolic symbiosis between the stromal and cancer cells, influencing growth, invasiveness and metastatic capacity. Previous research, combined with our findings, emphasizes the role of Cpt1a in shaping the immune landscape within tumor lesions[81]. Tumors lacking Cpt1a exhibit an increased presence of anti-inflammatory monocytes and anti-tumorigenic macrophages (M1-differentiated), which correlated with improved patient outcomes[82]. We replicated these findings while providing additional data arguing that Cpt1a targeting elicits broader anti-tumor immunity to facilitate responses to ErbB2 inhibition (Fig. 7d–g, and Supplementary Fig. 12). Specifically, the enhanced infiltration of cytotoxic T cells, Natural Killer cells and anti-tumor macrophages (M1 macrophages) in 7.16.4 mAb/Cpt1a-inhibited tumors aligns with their role in tumor elimination, including ADCC engagement, in ErbB2+ breast cancer, where their presence may correlate with favorable outcomes[83–86]. Ongoing clinical trials investigating strategies to improve anti-ErbB2 mAb therapy, such as ex vivo activated allogenic NK cells or macrophages engineered to express an anti-ErbB2 chimeric antigen receptor (CAR-M), may benefit from the enhanced anti-tumor immune responses induced by Cpt1a inhibition, potentially improving outcomes for tumors evading ADCC and resistant to Trastuzumab[87,88].

While our findings with ErbB2+ GEMMs validated the effects of FAO inhibition using etomoxir, a known CPT1-specific inhibitor, concerns about off-target effects at higher doses call for the development of more specific and effective Cpt1a inhibitors (Fig. 2b and Supplementary Fig. 2a)[89]. Promising candidates, such as ST1326, have demonstrated potential in hematological malignancies, potentially enhancing the therapeutic potential of targeting FAO in breast cancer[90]. In conclusion, our study provides valuable insights into the significance of targeting FAO, particularly Cpt1a, in HER2-positive breast cancer. The additive approach of combining Cpt1a inhibition with the ketogenic diet or anti-HER2 mAb therapy demonstrates promising results, including reduced tumor growth, proliferation, and metastasis (Fig. 8). By exploiting metabolic vulnerabilities, these findings offer opportunities to improve responses and combat resistance, leading to better outcomes for patients with aggressive HER2-positive primary and metastatic disease.

## Methods

### Ethics statement

All experiments performed in this study comply with ethical regulations for research using animals and data derived from human subjects. Animal studies were carried out under an Animal Use Protocol (#MCGL-5518) approved by the McGill University Downtown Campus Facility Animal Care Committee (FACC), a branch of the McGill University Animal Care Committee (UACC) and were conducted in compliance with McGill University and Canadian Council on Animal Care (CCAC) ethical guidelines. Mice were housed in autoclaved cages with *ad libitum* access to food and water, as well as appropriate and sufficient nesting and bedding materials. Mice had a 12 h cycle of light and darkness. Mouse cages were kept on ventilated racks, kept at a temperature range of 20–24 °C and a relative humidity of 45–65%.

### Animal models

For all mammary tumor studies, palpation to detect tumor onset and caliper measurements of tumors was performed twice weekly. In all experiments, mice were euthanized according to approved facility protocols at the institutionally approved tumor volume endpoint of 2.5 cm³ for a single mass or a total of 5 cm³ for multifocal tumors. The maximal tumor volume endpoint was not exceeded for any experiments in this study. As this study pertains to female breast cancer, sex as a biological variable was not considered in the study design and only female mice were used in experiments.

Transgenic models: MMTV-NIC and *Cpt1a* conditional mice were bred on a pure FVB/N background (Charles River, FVB/Ncrl; Strain code: 207) and were genotyped by PCR (see Supplementary Table S1 for primers). *Cpt1a* conditional mice were a gift from Russell Jones of the Van Andel Institute (Michigan, United States). All transgenic mice were generated through an in-house breeding program. Female littermates were group housed under specific pathogen-free conditions with a 12 h day/night cycle and *ad libitum* access to food, standard rodent chow (Teklad, Inotiv, 2920X) and water. Cohorts of female MMTV-NIC mice carrying wild-type or conditional alleles of *Cpt1a* (n = 20 per genotype) were monitored for mammary tumor formation by twice weekly palpation. Tumor monitoring began on 8-week-old and continued until the end of the experiments at 10 months. Once detected, tumors were measured weekly using calipers until they had reached a volume of 2.5 cm³ in size for a single mass or a total volume of 5 cm³ for multifocal tumors, at which point mice were euthanized in accordance with approved facility protocols.

In vivo ketogenic diet therapeutic studies: $1 \times 10^6$ cells were suspended in 30 µl of PBS and injected into the mammary fat pads of 8-week-old female FVB/N mice. Mice were kept on a standard rodent diet (Teklad, Inotiv, 2920X) for 4 days to allow appropriate recovery, after which half of each group were placed on the *ad libitum* Long-Chain Ketogenic diet (LCKD) (BioServ, F3666), mainly composed of lard, butter and corn oil, or Medium-Chain Ketogenic Diet (MCKD) (Bioserv, F10595), mainly composed of coconut oil. The compositions of the diets used in this study are detailed in Supplementary Table S2. Group sizes for the experiments were n = 6 per condition. Starting on day 8, mice were monitored twice weekly for mammary tumor formation by palpation, and weekly for blood glucose and ketone measurements. Tumors growth was measured using calipers as described above. At tumor endpoint, plasma was obtained by centrifuging blood samples at $1000 \times g$ for 15 min at 4 °C. To control for levels of ketosis, body weight, and colorimetric enzymatic kits were used to measure plasma glucose (Abcam, ab272532) and ketone (3-OHB) (ab83390) levels following the manufacturer's instructions.

Orthotopic allografts: $1 \times 10^6$ cells were suspended in 30 µl of PBS and injected into the mammary fat pads of 8-week-old female FVB/N mice.

In vivo therapeutic studies: Female mice were randomly assigned to treatment groups (n = 10 mice per treatment group) and monitored for tumor growth by twice-weekly palpation. Treatments began once tumors reached a size of $5 \times 5$ mm (approximately 65 mm³), with termination of the experiment when all isotype-matched controls had reached the end-point. 7.16.4 mAb (Bio X Cell, BE0277) and a mouse

isotype-matched control antibody IgG1 mAb (Bio X Cell, BE0083) was administered in doses of 100μg via intraperitoneal injection, 2 times a week. Mice were weighed twice-weekly, and doses were adjusted according to bodyweight. Tumor growth was measured by twice-weekly caliper measurements. Drug administration, tumor measurements and data analysis were performed by separate individuals who were blinded with respect to the treatment of each mouse.

### Histology and immunostaining

Tissues were fixed for 24 h in 10% neutral buffered formalin, paraffin embedded and sectioned at 4μm. Sections were stained with H&E or processed further as indicated.

**Lung metastasis.** Three H&E stained 10 um step sections per sample were scanned using an Aperio-XT slide scanner (Leica Biosystems) and analyzed using Imagescope software (Leica Biosystems).

**Immunohistochemistry.** For IHC, sections were deparaffinized and blocked as above and endogenous peroxidase activity was quenched by incubation in 3% hydrogen peroxide for 20 min. Sections were incubated with primary antibody overnight at 4 °C, washed three times in PBS and then incubated with ImmPRESS HRP polymer reagents to detect rabbit (Vector Elite, MP-7401) according to the manufacturer's instructions. After three further washes in PBS, IHC staining was visualized using the SignalStain DAB Substrate kit (Cell Signaling, 8059 S) according to the manufacturer's instructions. Sections were then counterstained with hematoxylin, dehydrated, and mounted with Clearmount (Invitrogen, 10058832). Images were acquired using an Aperio-XT slide scanner and analyzed using a nuclear staining algorithm in the associated software (Aperio Technologies).

**Immunofluorescence.** Sections were deparaffinized and dehydrated using 3 min incubations in xylene and decreasing concentrations of ethanol. Antigen retrieval was then performed in 10 mM EDTA (pH 9.0, Vector Laboratories, #H-3301) using a pressure cooker. Sections were incubated with 3% hydrogen peroxide and blocked in 10% Power Block (BioGenex, HK083) in TBS for 10 min at room temperature. Subsequently, sections were incubated with primary antibody in 2% (wt/vol) BSA in PBS at 4 °C overnight. Secondary antibody incubation was performed with ImmPRESS HRP polymer detection kit (Vector Laboratories, VECTMP745250, VECTMP740150, VECTMP740450) for 30 min and tyramide signal amplification substrates (Akoya Biosciences, OP-001001, OP-001003, OP-001004; Perkin Elmer, FP1489) were then added for 10 min at room temperature. Between each step, three washes in PBS were performed. Slides were incubated with DAPI (Thermo Fischer Scientific, D1306) for 10 min at room temperature, washed 3 times in water and mounted in ImmuMount (Thermo Fischer Scientific, 9990412). Imaging was performed using a Zeiss LSM800 confocal microscope or Zeiss AxioScan Z1 digital slide scanner, and staining was analyzed using HALO software (v3.5.3577, Indica Labs) using the algorithm 'Fluorescent intensity'. The entire tissue section was quantified for all staining experiments. Primary antibodies are detailed in Supplementary Table S3.

### Primary cell cultures and cell lines

Mammary tumors harvested from female MMTV-NIC mice at 8 weeks post-palpation[91], were dissociated in collagenase B (Roche, 11088831001)/Dispase II (Roche, 4942078001) (2.4 mg/ml each) for 2 h at 37 °C. The dissociated cells were washed three times with 1 mM EDTA in PBS and plated in Complete Media. Complete media consisted of DMEM (Wisent, 319-005-CL) supplemented with 2% FBS (Wisent, 080-150), 5 ng/ml EGF (Winsent, 511-110-UM), 1 μg/ml Hydrocortisone (Sigma, H4001), 5 μg/ml Insulin (Wisent, 511-016-UG), 35 μg/ml Bovine Pituitary Extract (BPE – Hammond CellTech, 1078-NZ), 25 μg/ml Amphotericin B (Wisent, 450-105-QL), 50 μg/ml Gentamycin Sulfate

(Wisent, 450-135-XL) and 50μg/ml Penicillin/Streptomycin (Wisent, 450-200-EL). The cells were cultured in a humidified, 5% $CO_2$, 37 °C incubator in Complete Media. Authentication of cells was performed using by PCR-based genotyping (See Supplementary Table S1 for primers) and immunoblotting to detect Cpt1a expression. The human cell line, 293 T (CRL-3216) was purchased from ATCC, used at early passage, and maintained in DMEM supplemented with 10% FBS. Regular biweekly testing of cells for mycoplasma using the MycoAlert Kit (Lonza, LT07-118) confirmed that all cell lines used in this study were negative for mycoplasma contamination.

### In vitro migration and invasion assays

Boyden chambers, BD Falcon Cell Culture Inserts, (8μm pore, BD Falcon, 353097) were prepared by coating the upper chamber with 50μL of DMEM (for migration) or DMEM containing 5% growth factor-reduced Matrigel (for invasion) (Corning, 356234). These chambers were placed into 24-well plates (Falcon, 353047) containing 1 mL of complete media in the lower chamber and incubated for 1 h at 37 °C. Cells (150,000 cells/500μL in DMEM) were seeded in triplicate into the upper chamber and incubated for 24 h at 37 °C with 5% $CO_2$. The boyden chambers were then fixed in 10% neutral-buffered formalin for 30 min, washed 3 times with water and counterstained with crystal violet for 30 min. Non-migrating cells from the upper chamber were wiped away with a cotton swab. Images were acquired using an Axio-Zoom V16 (Zeiss) and quantified for positive pixel area using ImageJ (v1.53 u, NIH).

### Immunofluorescent staining of mitochondria

Cells were fixed in 2% paraformaldehyde (PFA) (Sigma, 158127) in PBS at 37 °C for 10 min, then washed 3 times with PBS. After 3 washes in PBS, cells were permeabilized in 0.1% Saponin (Sigma, 47036) in PBS, followed by 3 washes in PBS. Then cells were blocked with 2% bovine serum albumin (BSA) in PBS, followed by incubation with primary antibodies, HSP60 and TOMM20 (detailed in Supplementary Table S3), in 2% BSA in PBS, for 1 h at room temperature. After 3 washes with PBS, cells were incubated with Alexa fluor 488 or 555 secondary antibodies (1:1000) (Thermo Fischer Scientific) as detailed in Supplementary Table S3 for 1 h at room temperature. After 3 washes in PBS, cells were incubated with DAPI (Thermo Fischer Scientific, D1306) for 10 min at room temperature, washed 3 times in water and coverslips were mounted in ImmuMount (Thermo Fischer Scientific, 9990412). Stained cells were imaged using the 60X objective lens on the Zeiss LSM800 confocal microscope with appropriate lasers. For mitochondrial morphology analysis, images from cells labeled for HSP60 and TOMM20 were obtained using the 60X objective and stacked in the same condition of gain, laser intensities and exposure time. Images were then compiled as "Max projection" and analyzed using the ImageJ software (v1.53 u, NIH).

### Quantitative reverse transcriptase-polymerase chain reaction

Total RNA was extracted from cultured cells or flash-frozen mammary tumors using the RNeasy Mini Kit (Qiagen, 74106). mRNA was reverse-transcribed into cDNA using the ProtoScript First Strand cDNA Synthesis Kit (New England Biolabs, E6300). Real-time quantitative PCR (qRT-PCR) was performed using the LightCycler 480 SYBR Green 1 MasterMix (Roche, 04887352001), run on the LightCycler 480 instrument (Roche) and analyzed using the corresponding software (Light-Cycler 480 SW, v1.5.1.62). Each sample was run in triplicate and normalized to *Actb* as a control. qRT-PCR Primer sequences are in Supplementary Table S4.

### Transcriptomic analysis

RNA was isolated as above from two (NIC/Cpt1a$^{+/+}$) or four (NIC/Cpt1a$^{L/L}$) independent tumor-derived cell lines in duplicate, and quality was assessed using a Nanodrop 2000 (Thermo Fisher

Scientific, ND2000CLAPTOP). RNA was sequenced and analyzed by Novogene[92]. Briefly, the reference genome mm10 (mouse) index was obtained from NCBI/ UCSC/ Ensembl. Clean reads were processed by taking the raw reads and removing reads containing adapters, with more than 0.1% of undetermined bases and were of low quality. Using STAR (v2.5), clean reads were mapped directly to the reference genome. Read counts were performed using HTSeq v0.6.1 and transcript abundance was determined using Fragments per kilobase of transcript sequence per million base pairs sequenced (FPKM) which accounts for sequencing depth and gene length. Differential expression analysis was performed using the DEGseq2 R package (2_1.6.3) with $p$-values adjusted using Benjamini and Hochberg's method. Genes with adjusted $p$-values under 0.05 were assigned as differentially expressed. FPKM levels were utilized to evaluate correlation differences and plotted using unsupervised hierarchal clustering, self-organization mapping (SOM) and kmeans. Analysis of transcriptional regulation in differentially expressed genes was performed using Enrichr[93].

### RNA FISH
RNAscope In-situ hybridization was performed on paraffin-embedded tumor sections using RNAscope® 2.5 HD Assay-RED kit (ACD, 322360) according to the manufacturer's protocol. The following probe was used: *IFNγ* (ACD, 311391). For Supplementary Fig. 12e, f, this protocol was followed with fluorescent IHC.

### Intracellular ROS detection
30,000 cells were seeded on coverslips in a 24-well plate. After 24 h, cells were incubated at a final concentration of 5 µM CellROX© Orange Reagent (Thermo Fischer Scientific, C10443) for 30 min at 37 °C, after which they were washed 3 times with PBS. Coverslips were then incubated with DAPI (Thermo Fisher Scientific, D1306) for 10 min at room temperature, washed 3 times in water and mounted in Immu-Mount (Thermo Fischer Scientific, 9990412). Images were acquired using an EVOS FL Microscope (Thermo Fischer Scientific, 4471136) and analyzed (positive pixel) using ImageJ (v1.53 u, NIH).

### Protein extraction and immunoblotting
Excised tumor tissue was immediately flash-frozen and crushed using a mortar and pestle under liquid nitrogen. The samples were allowed to thaw briefly and then lysed in ice-cold RIPA buffer (Tris-HCl 50 mM, pH 7.4, sodium chloride 150 mM, 1% Nonidet P-40, 1% sodium deoxycholate, 0.1% SDS, 2 mM EDTA, 0.5 mM AEBSF (Santa Cruz, sc-202041), 25 mM beta-glycerophosphate (Sigma, G5422), 1 mM sodium orthovanadate (BioShop, SOV664), and 10 mM sodium fluoride (Sigma, S7920)) by rotation at 4 °C for 30 min. Cultured cells were also lysed on ice in RIPA buffer. After lysis, lysates were cleared by centrifugation at 4 °C, $15,000 \times g$ for 10 min. Protein concentrations were determined using the Bradford assay (Bio-Rad, 5000006) and 40 µg of total protein was analyzed by SDS-PAGE followed by fluorescent immunoblotting using the Li-COR Odyssey system. Quantification was performed using the associated software, Image Studio Lite (v5.2.1, Li-COR Biosciences). Primary and secondary antibodies are detailed in Supplementary Table S2. (See Source Data for uncropped images of all immunoblots).

### Plasmids, lentiviral production and transduction
Sigma MISSION pLKO.1 constructs harboring shRNAs against: mouse Nrf2 (*Nfe2l2*) (Clone IDs: shNrf2-1, TRCN0000012128; shNrf2-2, TRCN0000012132; shNrf2-3, TRCN0000054659), mouse Keap1 (Clone ID: shKeap1-1, TRCN0000099445; shKeap1-2, TRCN0000099 446; shKeap1-3, TRCN000295016), mouse Brp44 (Mpc2) (Clone ID: shBrp44-1/shMpc2-1, TRCN0000195773; shBrp44-2/shMpc2-2, TRCN 0000241205), the non-mammalian target Luciferase (Clone ID: TRCN0000072259) and human MPC2 ORF construct (Clone ID:

TRCN0000473419) were obtained from the Mission TRC library (Sigma) provided by the McGill Platform for Cellular Perturbation (MPCP) at McGill University. Lentiviruses being shRNAs or ORFs were produced in 293 T cells (ATCC) co-transfected with the envelope plasmid, pMD2.G (Addgene, 12259) and packaging plasmid, psPAX2 (Addgene, 12260), using Lipofectamine 3000 (Invitrogen, L3000075) according to manufacturer's protocol. Virus-containing media were collected at 24 h and 48 h post-transfection, filtered through a 0.45 µm filter and stored at −80 °C. Cells were transduced with lentiviruses in the presence of 10 µg/mL polybrene (Sigma, 107689), selected and maintained in Complete media containing 2 µg/mL puromycin (Bio-shop, PUR333) for 2-3 days.

### Analysis of metabolite levels in conditioned media
Glucose and lactate levels in conditioned media were measured using a Flux Bioanalyzer (NOVA Biomedical) according to the manufacturer's instructions. For each condition tested, 2 ml of conditioned culture medium from $1 \times 10^6$ cells cultured in 6-well plates (Nunc) for 24 h was used. Detached cells and debris were removed by centrifuging media at $15,000 \times g$ for 10 min, 4 °C prior to use.

### Preparation of fatty acid-BSA conjugates and nutrient supplementation assay
Low glucose media, 2.5 mM and 1 mM glucose, was prepared in DMEM ([−] D-glucose, [−] Sodium Pyruvate) (Wisent, 319-062-CL) supplemented with 110 mg/L Sodium Pyruvate (Wisent, 600-110-EL) and D-glucose (BioShop, #GLU501) to achieve desired glucose concentration. Standard DMEM ([+] D-glucose, [+] Sodium Pyruvate) was used for 25 mM glucose media. Palmitic acid (#P0500), Caprylic Acid (#C5028), Capric Acid (#C4151), Lauric Acid (#L9755) and 3-hydroxybutyrate (3-OHB) (#166898) were purchased from Sigma Aldrich. Palmitic, caprylic, capric and lauric acid and [U-13C]-palmitate were conjugated with fatty acid free, low endotoxin bovine serum albumin (BSA) (Sigma Aldrich, #A1595). Carnitine (Sigma Aldrich, #C0283) was also included in fatty acid-supplemented conditions. Palmitate, caprylate, caprate, laurate and 3-OHB were prepared to the desired concentration in DMEM ([+] D-glucose, [+] Sodium Pyruvate). Incucyte Proliferation Assay was conducted to assess the effects of varied nutrient supplementation on percentage confluence of NIC cells as described previously.

### Respirometry and extracellular acidification measurements
Oxygen consumption and extracellular acidification rates were measured using an Xfe96 Extracellular Flux Analyzer (Seahorse Bioscience, North Billerica, MA) using the manufacturer's established protocols. In brief, cells were plated overnight in Seahorse 96-well plates at $1 \times 10^4$ per well in 80 µL of complete media. Cells were washed three times in non-buffered DMEM containing 25 mM glucose and 2 mM glutamine and incubated in this medium in a $CO_2$-free incubator at 37 °C for 2 h to allow for temperature and pH equilibration before loading into the Xfe96 apparatus. In addition to basal measurements, the Mito Stress Test assay (Seahorse Bioscience, 103015-100) was performed according to manufacturer's instructions. Xfe assays consisted of sequential mix (3 min), pause (3 min), and measurement (5 min) cycles, allowing for determination of OCR/ECAR every 10 min. At the end of the assay, the IncuCyte S3 system (ESSEN BioSciences, Ann Arbor, MI, USA) was used for live cell counting at 10x magnification. Cell counts were determined using the IncuCyte S3 Analysis software (v2019A, ESSEN BioSciences). Data was normalized to cell number (20,000 cells) and analyzed using Wave software (v2.6.0.31, Seahorse Bioscience).

### Measurement of exogenous and endogenous fatty acid oxidation
FAO of exogenous long-chain (Palmitate) and medium chain (Laurate) fatty acids was assessed using the Seahorse Xfe96 extracellular flux

analyzer, in accordance with the manufacturer's instructions. Briefly, NIC cells seeded on Seahorse 96-well plates were cultured in substrate-limited DMEM ([-] D-glucose, [-] Sodium Pyruvate (Wisent, 319-062-CL)) with 0.5 mM D-glucose (BioShop #GLU501), 1 mM GlutaMAX (Sigma Aldrich, #G7021), 0.5 mM Carnitine (Sigma Aldrich, #C0283) and 1% FBS (Wisent, 080-150). On the day of the assay, substrate-limited medium was exchanged for FAO medium (111 mM NaCl, 4.7 mM KCL, 1.25 CaCl$_2$, 2 mM MgSO$_4$ and 1.2 mM NaH$_2$PO$_4$ supplemented with 2.5 mM D-glucose, 0.5 mM Carnitine, 5 mM HEPES (Sigma Aldrich, #H0887)). Prior to the initiating the XF assay, cells were treated with either DMSO or 10 μM etomoxir for 15 min. BSA control substrate and XF-Palmitate-BSA FAO (Seahorse Bioscience, #102720-100) or Laurate-BSA FAO substrate (conjugated by previously described methods) were added just before the start of the assay. The MitoStress test was performed, and cell counts were determined using the IncuCyte S3 analysis software (v2019A, ESSEN BioSciences). Data normalization was conducted to 20,000 cells using the Wave software (v2.6.0.31, Seahorse Bioscience).

## Metabolomics and gas chromatography/mass spectrometry analysis

Metabolic profiling and isotope tracing analyses using $^{13}$C-glucose and $^{13}$C-palmitate were performed at the Metabolomics Innovation Resource, McGill University. Briefly, Cpt1a-proficient and -deficient NIC cells ($n$ = 3 cell lines per genotype, analyzed in triplicate) were cultured in unlabeled DMEM supplemented with 2% dialyzed FBS (Wisent, 080-910) in 6-cm dishes (Nunc) for 48 h. The media was then replaced with glucose-free DMEM supplemented with 2% dialyzed FBS and 25 mM [U-$^{13}$C]-glucose (Cambridge Isotope Laboratories, CLM-1396, 99% atom $^{13}$C) for either 30 min or 2 h, or [U-$^{13}$C]-palmitate (Cambridge Isotope Laboratories, CLM-409, 99% atom $^{13}$C) for 24 h. Additionally, some dishes were kept in unlabeled media as controls. Cells were washed three times in ice-cold saline solution (NaCl, 0.9 g/L), and water-soluble metabolites were extracted in 80% methanol (GC/MS grade). After two 10-minute rounds of sonication (30 s on/ 30 s off at high intensity) on slurry ice using the Bioruptor UCD-200 sonicator, the homogenates were centrifuged at 15,000 × $g$ for 10 min at 4 °C. Supernatants were collected and an internal standard, 800 ng Myristic acid-D27, was added to each sample. Samples dried by vacuum centrifugation (CentriVap Concentrator; Labconco, KS, USA) overnight at −1 °C were resuspended in 30 μl of 10 mg/mL methoxyamine hydrochloride in anhydrous pyridine and incubated for 30 min at room temperature. Samples were then transferred to GC−MS autoinjector vials containing 70 μl N-(tert-butyldimethylsilyl)-N-methyltri-fluoroacetamide (MTBSTFA) derivatization reagent and incubated at 70 °C for 1 h. A blank sample, composed of 30 μl of 10 mg/mL methoxyamine-HCl pyridine and 70 μl of MTBSTFA, was also prepared. A volume of 1 μl of sample was injected splitless with an inlet temperature of 280 °C into the GC-MS instrument, Agilent 5975 C. Metabolites were resolved by separation on a DB-5MS + DG (30 m × 250 μm × 0.25 μm) capillary column (Agilent Technologies, CA, USA). Helium was used as the carrier gas with a flow rate such that Myristic-D27 acid eluted at approximately 18 min. The quadrupole was set at 150 °C, the source at 230 °C, and the GC/MS interface at 320 °C. The oven program started at 60 °C, held for 1 min, then increased at a rate of 10 °C per minute until 320 °C. Bake-out was at 320 °C for 9 min. Metabolites were ionized by electron impact at 70 eV. All samples were injected using scan (50−1000 m/z) and selected ion monitoring (SIM) mode. In all experiments, Cpt1a-proficient NIC cells were used as controls. The sample preparation and data collection order for biological and technical replicates was randomized. All metabolites described in this study were validated against authenticated standards to confirm mass spectra and retention times. The relative amount of each metabolite was determined from the integration of ion intensities and normalized to the number of cells extracted using MassHunter Quant software (v12.0.893.1, Agilent Technologies) according to published protocols[94]. Mass isotopomer distribution analysis was determined using a custom in-house algorithm developed at McGill University (McGuirk)[95,96].

## Neutral lipid staining and uptake of BODIPY using flow cytometry

Cells were treated with 2 μM of cell membrane-permeant fluorophore, BODIPY 493/503 (4,4-Difluoro-1,3,5,7,8-Pentamethyl-4-Bora-3a,4a-Diaza-s-Indacene) (Thermo Fischer Scientific, #D3922) or BODIPY FL C16 (4,4-Difluoro-5,7-Dimethyl-4-Bora-3a,4a-Diaza-s-Indacene-3-Hex-adecanoic Acid) (Thermo Fischer Scientific, #D3821) for 30 min at 37 °C to stain for lipid droplets. Samples were washed with PBS to remove staining solution, trypsinized to generate single cell suspension and centrifuged at 800 × $g$ for 3 min. Subsequently, cells were washed with FACS buffer (1X PBS with 2 mM EDTA and 2% FBS), passed through a 70 μm strainer and stained with viability dye eFluor$^{YM}$ 506 (eBioscience, 65-00866-18) for 10 min on ice. A minimum of 100,000 events per sample was acquired using a LSR Fortessa 5 L Flow Cytometer (BD Bioscience, H649225B4076) with FACSDiva Software (Version 8) (BD Biosciences) in slow rate mode to avoid doublets. Cell populations were gated as shown in Supplementary Fig. 2d. Data were analyzed with FlowJo Software (v10.10_CL, Ashland, OR, USA). Cell debris and aggregates were excluded from the analysis using pulse processing SSC-H vs SSC-W.

## Intracellular nucleotide measurement assays

Cells were seeded onto 10 cm dishes to measure intracellular Nucleotide (ATP, FAD, NAD$^+$, NADH, NADP$^+$, NADPH) levels. The live cell numbers were counted prior to the assay with trypan blue staining using Cellometer Auto T4 Automated Cell Counter (Nexcelom), and 1 × 10$^6$ (ATP, FAD), 2 × 10$^6$ (NAD$^+$, NADH) and 4 × 10$^6$ (NADP$^+$, NADPH) cells were subjected to ATP (Abcam, ab83355), FAD (Abcam, ab204710), NAD$^+$ and NADH (NAD$^+$/NADH; Abcam, ab65348), and NADP$^+$ and NADPH (NADP$^+$/ NADPH; Abcam, ab65349) assay kits according to the manufacturer's instructions. Nucleotides levels were measured with microplate reader spectrophotometer at 570 nm (ATP and FAD) and 450 nm (NAD$^+$/NADH and NADP$^+$/NADPH). Nucleotide levels were determined from a standard calibration curve and nucleotide concentrations were derived from the sample volume used for the measurement.

## In vitro drug preparation and Incucyte cell proliferation assays

Etomoxir (MedChemExpress, HY-50202), N-Acetylcysteine (Med-ChemExpress, HY-B0215), UK-5099 (MedChemExpress, HY-15475), Brusatol (MedChemExpress, HY-19543) and ML385 (MedChemExpress, HY-100523) were dissolved in DMSO in vitro as described. Cells were treated at the indicated concentrations for 24 h for immuno-blotting, qRT-PCR, OCR/ECAR measurements and mitochondrial staining experiments. For proliferation assays, 5000 cells per well were seeded in triplicate or quadruplicate in 96-well optical-bottom plates (Nunc, 167008). After 24 h of seeding, drugs or vehicle controls were added, and live cell imaging was performed using the IncuCyte S3 system (ESSEN BioSciences, Ann Arbor, MI, USA) at 10x magnification every 6 h over a period of 96 h (2 images per well per timepoint, 17 total timepoints). Percentage confluence was determined using the IncuCyte S3 Analysis software (v2019A, ESSEN BioSceinces).

## Analysis of transcriptomic data from publicly available datasets

Kaplan−Meier curves depicting overall survival (OS) based on *CPT1A* mRNA expression in breast cancer and HER2+ breast cancer patients were generated using the Gene Expression Profiling Interactive Analysis database (GEPIA2) web-based tool (http://gepia2.cancer-pku.cn/) from Breast invasive carcinoma (BRCA) TCGA/GTEx dataset ($n$ = 1085)[97]. To assess the prognostic value of CPT1A, patient samples

were stratified into two groups, high and low, based on median cut-offs. The two patient cohorts were compared using Kaplan–Meier survival plots, and the hazard ratios, calculated using the Cox proportional-hazard model with 95% confidence intervals (CIs), and log-rank $p$ values were displayed on the figure. $p$-values less than 0.05 were considered significant. Receiver Operating Characteristic (ROC) plots were used to assess whether *CPT1A* gene expression could serve as a biomarker for sensitivity or resistance to Trastuzumab in breast cancer. These ROC plots were generated from ROC plotter (https://rocplot.org) using Breast Cancer datasets identified in the Gene Expression Omnibus (GEO) (http://www.ncbi.nlm.nih.gov/gds). The GEO platform IDs used included, "GPL96" (for HG-U133A), "GPL570" (for HG-U133 Plus 2.0), "GPL571" (for HU-U133A_2), the keywords "breast", "cancer" and "therapy" and datasets with fewer than 30 samples were excluded ($n = 3104$)[98]. These datasets were selected as they are widely used and utilize the same probes for measuring the same genes. "ROC Plotter for breast cancer" tool was selected and CPT1A was entered as the gene symbol. ROC plots were created for both complete pathological response (cPR) and relapse-free survival (RFS) to Anti-HER2 therapy, Trastuzumab.

## Statistics and reproducibility
GraphPad Prism 9.0 software was used to generate graphs and statistical analyses and compiled on Adobe Illustrator (v25.2.3). Statistical significance was determined by unpaired, two-tailed Student's $t$-tests, one-way ANOVA with Tukey's post-hoc tests for multiple comparisons, and Kaplan–Meier analysis with logrank tests (Mantel–Haenszel), as appropriate. Throughout the study, data are presented as the mean ± SD, with statistical significance defined at a $p$-value of less than 0.05 ($p < 0.05$). The methods for statistical tests, sample sizes ($n$), exact $p$-values (for *$p < 0.05$, **$p < 0.01$, ***$p < 0.001$, ****$p < 0.0001$) and definition of error bars are indicated in the figure legends. Statistical comparisons not indicated on the figure were found to be insignificant. No power analysis or statistical method was used to calculate the sample size; they were determined based on previous experience. $p$-values from statistical tests were used to assess statistical significance and appropriateness of sample sizes. No data were excluded from the analyses. All experiments were reproduced in at least two independent experiments using the indicated biological and technical replicates unless otherwise specified in the figure legends. All immunoblots and images presented are representative of these independent experiments. For in vivo experiments, mice were randomly allocated to treatment groups, and investigators performing drug treatments and tumor measurements were blinded to the allocation.

## Reporting summary
Further information on research design is available in the Nature Portfolio Reporting Summary linked to this article.

## Data availability
The RNA-Seq data generated in this study have been deposited in the Gene Expression Omnibus (GEO) database under accession code GSE254622. The raw metabolomics data in this study were generated at the Metabolomics Innovation Resource Facility and have been deposited in the Mendeley Data database under accession code DOI: 10.17632/ddxckj7s5b.3 [https://data.mendeley.com/datasets/ddxckj7s5b/3]. All findings from this study are available within the article, supplementary information, and source data files. Source data are provided with this paper.

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

## Acknowledgements

We thank members of the Muller lab for their suggestions and support and acknowledge the Goodman Cancer Institute Histology Core, McGill Platform for Cellular Perturbation, Bioinformatics Core Technology Platform, Metabolomics Innovation Resource, Advanced BioImaging Facility and the McGill Comparative Medicine & Animal Resources Center for their technical assistance. Our research was funded by the Canadian Institutes of Health Research (CIHR) (CIHR PJT-186155), Terry Fox Research Institute (TFRI) Program Project Grant #1091 and the Canada Research Chair in Molecular Oncology (CRC – 950231033 ×216779) (W.J.M.), CURE Foundation Fellowship in Breast Cancer Research and The Rosalind Goodman Commemorative Scholarship (I.N.), the US Department of Defense Congressionally Directed Medical Research Programs, Breast Cancer Research Program W81XWH-11-1-0046 (H.W.S.).

## Author contributions

I.N.: conceptualization, data curation, investigation, formal analysis, resources, methodology, writing – original draft, writing - review and editing, visualization and funding acquisition. L.J.: investigation, formal analysis and methodology. H.W.S.: conceptualization, resources, writing – review and editing. D.A.: investigation, formal analysis, methodology and visualization. V.P.: methodology and investigation. C.L.: methodology and investigation. A.P.: investigation, formal analysis, visualization. S.A.: methodology and investigation. V.S.-G.: investigation, formal analysis, visualization. W.J.M.: conceptualization, writing – original draft, writing – review and editing, funding acquisition and supervision.

## Competing interests

The authors declare no competing interests.
