## [Peer Review File · Nature Communications]

Targeting Fatty Acid Oxidation Enhances Response to HER2-targeted TherapyREVIEWER COMMENTS

Reviewer #1 (Remarks to the Author):

Review of NCOMMS-24-02043-T

In the manuscript by Nandi et al., titled "Targeting Fatty Acid Oxidation Enhances Response to HER2-targeted Therapy", the authors studied the impact of CPT1a in HER2-driven breast cancer. The study had a clear rationale, and the manuscript was concise and easy to follow. To begin, the authors demonstrated the importance of CPT1a to HER2-driven breast cancer using well-established in vivo mouse models. Notably, they demonstrated the importance of CPT1a to breast tumors over the other CPT1 isoforms. Next, they provided mechanistic details to suggest that in the absence of CPT1a, HER2+ breast cancer cells upregulate glucose uptake and metabolism through the TCA. Further, they found that loss of CPT1a induces oxidative stress and an NRF2-dependent transcriptional response, which drives the reliance on glucose in CPT1a-deficient cells. This mechanistic insight prompted them to explore how these findings could be translated into therapeutic approaches. While blocking NRF2 activity in vivo has its challenges, the authors eloquently suggested an alternative approach to instead deprive tumors of glucose using a ketogenic diet, which is low in carbohydrates (i.e., glucose). They demonstrated that while a ketogenic diet alone accelerates tumor growth, combining a ketogenic diet with the loss of CPT1a exacerbated a reduction in tumor growth. Finally, they extended their findings to the clinical setting by showing that loss of CPT1a enhances the response to HER2 inhibition therapy. This is an excellent study of high rigor and high innovation. Further, this study is of great interest to a broad audience. While additional experiments might add value to the manuscript, they are most likely outside the scope of the study. Thus, this Review has no major concerns with the manuscript in its current state, and only minor comments are provided.

Minor Comments:

1. Line 29 should delete "an" before "ErbB2+."
2. Line 400 should say "mitochondria" instead of "mitochondrial."
3. Line 581 should delete "an" before "restoring."
4. Supplementary Figure 7A could include a higher-resolution image of the heat map.
5. Line 625 should have "NRF2" in capitals. This nomenclature should be maintained throughout the manuscript.

Reviewer #2 (Remarks to the Author):

In this study, Nandi et al. generate a mouse breast cancer model in which CPT1A is knocked out in ErbB2-driven breast tumors. They show that CPT1A loss delays the growth and progression of these tumors. Using various metabolic approaches, they show that CPT1A-deficient breast cancer cells lose the capacity to oxidize fatty acids and adapt by increasing glucose oxidation, in a manner that may be dependent on NRF2 activation. Consistent with these observations, a ketogenic diet that reduces glucose availability and that may increase tumor dependence on fatty acids is able to further inhibit the growth of CPT1A-deficient tumors. In addition, CPT1A loss is also shown to potentially enhance the response of ErbB2 monoclonal antibody therapy.

The study is well done with a strong mouse model and good metabolic characterization of CPT1A loss. The effects of the ketogenic diet are particularly interesting, since the ketogenic diet enhances the growth of wild-type tumors but inhibits the growth of CPT1A-deficient tumors, suggesting the potential efficacy of combining a ketogenic diet with CPT1A inhibitors. The study would likely be strengthened the most by showing that CPT1A loss can be rescued by medium chain fatty acids both in vitro and in vivo, since the oxidation of these fatty acids should not be dependent on CPT1A. The effects of CPT1A loss and HER2 inhibition also appear to be more additive rather than synergistic.

Comments:

1. In Fig. 2G, why is succinate labeled at ~20%, but the downstream metabolites fumarate malate are labeled much less at 0.2% (2 orders of magnitude less)? Typically TCA metabolites are labeled at roughly similar percentages, as is shown in the glucose tracing experiment in Fig. S3D.

2. In Fig. 2H, decreased CD36 is consistent with overall decreased fatty acid uptake, and not just loss of fatty acid oxidation in CPT1A deficient cells. Can the authors directly measure and show decreased fatty acid uptake?
3. Can the phenotypes of CPT1A KO cells (decreased proliferation, migration/invasion, OCR, increased glucose oxidation, etc.) be rescued by supplementing cells with medium-chain FAs that can bypass CPT1?
4. In Fig. S4B, to address off-target effects of shRNA, the data would be strengthened with a shRNA-resistant rescue, and also showing the same phenotypes with the MPC inhibitor UK-5099.
5. Fig. S6: Is it the author's model that increased oxidative stress induction contribute to decreased proliferation of Cpt1a KO cells? In this case does NAC rescue the proliferation of CPT1A KO cells? However, the authors also argue that Nrf2 activation downstream of oxidative stress induction is responsible for the compensatory increased glucose oxidation- in this case, would NAC block Nrf2 activation and further impair growth, similar to the Nrf2 inhibitors? How does NAC influence GLUT1 expression, glucose uptake, and glucose utilization for lactate vs. TCA cycle?
6. Fig. 5B: it is stated that wild type cells are not dependent on NRF2 for glucose uptake, and yet NRF2 inhibition does inhibit glucose uptake to some extent in wild-type cells.
7. Fig. S8c: Why would CPT1a KO cells not respond to b-OHB when bOHB transport into the mitochondria and oxidation should not be dependent on CPT1a
8. Similar to point 3 above, would formulating a ketogenic diet with medium chain triglycerides rescue the effects of the KD + CPT1A KO combination?
9. Fig. 7D-F: There does not appear to be a synergy between CPT1A loss and HER2 inhibition- while the combination does lead to the slowest growing tumors, HER2 inhibition seems to have a stronger relative effect in WT tumors compared to CPT1A KO tumors. The combination is likely more additive than synergistic.
10. Abstract states that "combining a ketogenic diet with an anti-ErbB2 monoclonal antibody in the context of CPT1A deficiency significantly reduced tumor growth and increased survival". Unless I missed it, where is this data?

Reviewer #3 (Remarks to the Author):

In this report, Muller and colleagues carry out an analysis of the effects of loss of Cpt1a in HER2+ mammary tumors. In addition to examining growth, angiogenesis, and metastasis, they carry out an extensive analysis of the metabolic consequences of Cpt1a knockout. These studies not only revealed impaired mitochondrial function associated with an increased reliance on glycolysis and the TCA cycle for energy production, but also revealed a significant upregulation of oxidative stress and adaptive upregulation of the anti-oxidant program regulated by NRF2. The metabolic studies were quite thorough with metabolic tracing studies as well as analysis of effects of inhibitors to counteract adaptive programs. In addition, the authors explored several candidate therapeutic approaches and identified some candidate combinations. This report was impressive in the breadth and depth of analyses as well as inclusion of appropriate replicates and alternate approaches to investigate their findings.

These studies provide new insights into metabolic programs associated perturbation of fatty acid oxidation in HER2 tumors, and reveal candidate therapeutic strategies to target this pathway and adaptive programs that result from its inhibition.

There were only two minor issues that would improve the manuscript if addressed:

- 1) Figure 6d and e: . In Figure 6d, the data shows essentially no difference in weight of the HER2 tumors with or without Cpt1a KO at tumor endpoint; however figure 6e, which shows tumor volume, show ~ a 7- to 10- fold difference in tumor volume. Similarly, the Keto diet Cpt1a KO mice display a dramatic difference in tumor volume relative to the control WT mice. It is likely that this is an explanation for this but it isn't clear here and it is important to address this.
- 2) A keto diet was shown to increase tumor size and decrease survival in mice bearing wild-type ErbB2+ tumors, whereas "combining Cpt1a ablation with the ketogenic diet markedly decreased tumor growth and enhanced survival compared to all other conditions". This is an interesting finding which provided useful information to the authors comprehensive analysis of Cpt1a function in HER2 tumors; however they also discuss use of ketogenic diets together with CPT1A inhibition as therapeutic strategy. It is recommended to add a cautionary note about the use of any strategy in which one arm of combination therapy enhances tumor growth.
- 3) One question that comes from this analysis is how glutamine metabolism was affected by Cpt1a

deletion. It isn't critical for the report which is quite thorough, but if the authors looked at this in any of their metabolic studies, it would be informative to add this.

William J. Muller, Ph.D., FRSC

Rosalind and Morris Goodman Cancer Institute

Cancer Pavilion, McGill University

1160 Pine Ave. West, Room 516

Montreal, Québec, Canada H3A 1A3

May 10th, 2024

Response to Reviewer's Comments

We would like to thank the reviewers for taking the time to read our manuscript. We are very pleased to resubmit our revised manuscript (NCOMMS-24-02043-A) to Nature Communications. We are grateful for the positive overall responses to our work and note that all three of the Reviewers found this study to be “excellent” with “high rigor and innovation”, with the experiments described as “impressive in breadth and depth of analyses”, “well-done with a strong mouse model and characterization”, as well as “comprehensive”, “concise and easy to follow”. In addressing the reviewers’ comments, we have provided clarification where required, rectified minor inconsistencies, and strongly reinforced the conclusions of our study. We believe that the revised manuscript is significantly improved and is now suitable for publication in Nature Communications. Please find below a point-by-point response to the reviewer’s comments, which are also included in italics for ease of reference. New data have been integrated into the main or supplementary figures of the revised manuscript as indicated. We have tracked significant changes and additions to the text that reflect the Reviewers’ input. Other changes to the text were made for clarity and in adherence with Nature Communications editorial suggestions.

Reviewer #1:

In the manuscript by Nandi et al., titled “Targeting Fatty Acid Oxidation Enhances Response to HER2-targeted Therapy”, the authors studied the impact of CPT1a in HER2-driven breast cancer. The study had a clear rationale, and the manuscript was concise and easy to follow. To begin, the authors demonstrated the importance of CPT1a to HER2-driven breast cancer using well-established in vivo mouse models. Notably, they demonstrated the importance of CPT1a to breast tumors over the other CPT1 isoforms. Next, they provided mechanistic details to suggest that in the absence of CPT1a, HER2+ breast cancer cells upregulate glucose uptake and metabolism through the TCA. Further, they found that loss of CPT1a induces oxidative stress and an NRF2-dependent transcriptional response, which drives the reliance on glucose in CPT1a-deficient cells. This mechanistic insight prompted them to explore how these findings could be translated into therapeutic approaches. While blocking NRF2 activity in vivo has its challenges, the authors eloquently suggested an alternative approach to instead deprive tumors of glucose using a ketogenic diet, which is low in carbohydrates (i.e., glucose). They demonstrated that while a ketogenic diet alone accelerates tumor growth, combining a ketogenic diet with the loss of CPT1a exacerbated a reduction in tumor growth. Finally, they extended their findings to the clinical setting by showing that loss of CPT1a enhances the response to HER2 inhibition therapy. This is an excellent study of high rigor and high innovation. Further, this study is of great interest to a broad audience. While additional experiments might add value to the manuscript, they are most likely outside the scope of the study. Thus, this Review has no major concerns with the manuscript in its current state, and only minor comments are provided.

Minor Comments:

1. Line 29 should delete “an” before “ErbB2+.”
2. Line 400 should say “mitochondria” instead of “mitochondrial.”
3. Line 581 should delete “an” before “restoring.”
4. Supplementary Figure 7A could include a higher-resolution image of the heat map.
5. Line 625 should have “NRF2” in capitals. This nomenclature should be maintained throughout the manuscript.

We thank the reviewer for these suggestion and have incorporated and tracked all these changes in our revised Manuscript. Additionally, we appreciate that the magnification and resolution of the heat map shown in Supplementary Figure 7a of the original manuscript was insufficient. We have included higher resolution image of the heat map in the revised manuscript (now Supplementary Fig. 6a, following changes in Supplementary Figure file prompted by reviewer comments). Lastly, as the experiments for Figure 4 and Figure 5 are conducted in murine cell lines, we have changed all the protein names, from “NRF2” to “Nrf2”, from “KEAP1” to “Keap1”, “GLUT1” to “Glut1” and “MPC2” to “Mpc2” in both the Results, Figures, Figure Legends and Discussion. We thank the reviewer for bringing this to our attention.

Reviewer #2:

In this study, Nandi et al. generate a mouse breast cancer model in which CPT1A is knocked out in ErbB2-driven breast tumors. They show that CPT1A loss delays the growth and progression of these tumors. Using various metabolic approaches, they show that CPT1A-deficient breast cancer cells lose the capacity to oxidize fatty acids and adapt by increasing glucose oxidation, in a manner that may be dependent on NRF2 activation. Consistent with these observations, a ketogenic diet that reduces glucose availability and that may increase tumor dependence on fatty acids is able to further inhibit the growth of CPT1A-deficient tumors. In addition, CPT1A loss is also shown to potentially enhance the response of ErbB2 monoclonal antibody therapy.

The study is well done with a strong mouse model and good metabolic characterization of CPT1A loss. The effects of the ketogenic diet are particularly interesting, since the ketogenic diet enhances the growth of wild-type tumors but inhibits the growth of CPT1A-deficient tumors, suggesting the potential efficacy of combining a ketogenic diet with CPT1A inhibitors. The study would likely be strengthened the most by showing that CPT1A loss can be rescued by medium chain fatty acids both in vitro and in vivo, since the oxidation of these fatty acids should not be dependent on CPT1A. The effects of CPT1A loss and HER2 inhibition also appear to be more additive rather than synergistic.

Comments:

1. In Fig. 2G, why is succinate labeled at ~20%, but the downstream metabolites fumarate malate are labeled much less at 0.2% (2 orders of magnitude less)? Typically TCA metabolites are labeled at roughly similar percentages, as is shown in the glucose tracing experiment in Fig. S3D.

We thank the reviewer for bringing this discrepancy to our attention. Upon careful review, we have identified an oversight in the presentation of the labeling percentages in Fig. 2g. We acknowledge that succinate was labeled at approximately 20%, whereas downstream metabolites fumarate and malate were incorrectly labeled at a much lower labeling percentage. However, after carefully re-evaluating our data, we have confirmed that all metabolites, including succinate, fumarate, malate, and citrate, are indeed labeled in the range of approximately 15-60%. This discrepancy arose due to a mistake in the calculation of the fraction of relative abundance during the matrix correction of the values for fumarate and malate. We apologize for any confusion this may have caused and appreciate your diligence in bringing this to our attention. We have ensured that this error is corrected in the manuscript and the appropriate adjustments are made to accurately represent the labeling percentages of all metabolites in Fig. 2g. We thank you for your understanding.

2. In Fig. 2H, decreased CD36 is consistent with overall decreased fatty acid uptake, and not just loss of fatty acid oxidation in CPT1A deficient cells. Can the authors directly measure and show decreased fatty acid uptake?

We thank the reviewer for this insightful comment. As you can appreciate, there are various methods to measure fatty acid uptake, including assays with heavy-isotope labeled fatty acids, such as ¹³C₆-Palmitate or ¹⁴C₆-Oleate, fatty acid analogs conjugated to fluorescent probes, such as C1-BODIPY-C12 or BODIPY FL-C16, and measuring the expression of fatty acid transport proteins (FATP) and Cd36 (1-6). In our manuscript, to measure fatty acid uptake, we utilized labeled palmitate (¹³C₆-Palmitate) in the media and demonstrated its entry into the cell in wild-type HER2+ cells, where it undergoes fatty acid oxidation in the mitochondria, as evidenced by the labeling of TCA cycle intermediate metabolites (citrate, succinate, malate, and fumarate) (Fig. 2, f and g). This uptake and subsequent fatty acid oxidation into the TCA cycle was lost upon Cpt1a

ablation. Additionally, as you noted, we observed decreased expression of the fatty acid transporter Cd36 in cells lacking Cpt1a (Fig. 2h). However, we acknowledge that this alone does not conclusively demonstrate the inability of Cpt1a-deficient cells to uptake fatty acids into the cell, but rather indicates impaired uptake of long-chain fatty acids into the mitochondria for beta-oxidation. To address this directly, we utilized BODIPY 493/503 and the fluorescently labelled fatty acid analog, BODIPY FL-C16, to quantify fatty acid uptake and storage into Cpt1a-proficient and -deficient cells (6, 7). Using fluorescent cytometry, we observed significantly reduced BODIPY 493/503 and BODIPY FL-C16 staining in Cpt1a-deficient cells compared to wild-type HER2+ cells. The results showed that diminished fatty acid uptake and storage was observed in the absence of Cpt1a, respectively (Supplementary Fig. 2, d and e). We appreciate your valuable input, which has led us to further investigate fatty acid uptake in our study.

3. Can the phenotypes of CPT1A KO cells (decreased proliferation, migration/invasion, OCR, increased glucose oxidation, etc.) be rescued by supplementing cells with medium-chain FAs that can bypass CPT1?

We thank the reviewer for their suggestion. It is indeed true that Cpt1a primarily regulates the entry and oxidation of long-chain fatty acids into the mitochondria. Thus, in theory, medium-chain fatty acids (MCFAs) could bypass the Cpt1a blockade. To address this, we investigated whether supplementing Cpt1a-deficient cells with medium-chain fatty acids could reverse the observed phenotypes, including decreased proliferation, migration/invasion, oxygen consumption rate (OCR), and increased glucose uptake. Specifically, we focused on caprylic, capric and lauric acid, which are the predominant medium-chain fatty acids (MCFAs) found in medium-chain triglycerides (MCTs) (Caprylic acid – C8:0, Capric acid – C10:0, and lauric acid (C12:0))(8). Our analysis revealed that supplementation with all three MCFAs rescued proliferation in Cpt1a-deficient cells (Supplementary Fig. 11a). Given that lauric acid is the most prevalent MCFA found in coconut oil, the main component of the medium chain ketogenic diet, we opted to focus our subsequent characterization of Cpt1a-deficient cells using lauric acid. Remarkably, we observed a partial rescue in migration, invasion and oxygen consumption rate in Cpt1a-deficient cells upon supplementation with lauric acid (Supplementary Fig. 11, b and c). Despite the addition of lauric acid in the media, these cells still exhibited partial dependence on Nrf2 and glucose, as evidenced by elevated *Nrf2*, *Glut1* expression and glucose consumption compared to wild-type cells (Supplementary Fig. 11, d and e). However, the presence of lauric acid reduced their reliance on glucose when compared to Cpt1a-deficient cells without MCFAs supplementation (Supplementary Fig. 11, d and e). These results indicate that supplementation with MCFAs can indeed bypass Cpt1a. Overall, our findings suggest that Cpt1a-deficient cells can utilize both glucose and MCFAs as alternative fuel sources, highlighting the metabolic flexibility of these cells in adapting to changes in nutrient availability. We appreciate your insightful question, which has prompted us to further investigate the role of medium-chain fatty acids in rescuing the phenotypes associated with Cpt1a deficiency.

4. In Fig. S4B, to address off-target effects of shRNA, the data would be strengthened with a shRNA-resistant rescue, and also showing the same phenotypes with the MPC inhibitor UK-5099.

We thank the reviewer for the valuable feedback. To address concerns regarding potential off-target effects of the shRNA knockdown, we conducted several experiments to strengthen our data. Firstly, we assessed the effects of the MPC inhibitor UK-5099 on cell proliferation. We observed that while the MPC inhibitor minimally diminished proliferation of wild-type cells, it significantly impaired proliferation of Cpt1a-deficient cells, consistent with the effects observed upon silencing of *Mpc2* in these cells (Supplementary Fig. 4, d and e). These results indicate that pharmacological inhibition recapitulates the genetic ablation of *Mpc2*. Next, we overexpressed human MPC2, resistant to the targeting by shMpc2, in both Cpt1a-proficient and -deficient cells with shRNA knockdown of *Mpc2*, demonstrating that it partially rescues proliferation of Cpt1a-deficient cells with stable silencing of *Mpc2* (Supplementary Fig. 4f). As wild-type HER2+ cells are not dependent on *Mpc2*, they exhibited no significant alterations in proliferation upon human MPC2 re-expression. Moreover, we investigated the interplay between MPC inhibition and *Mpc2* re-expression in Cpt1a-null cells with *Mpc2* knockdown. We found that although re-expression of *Mpc2* partially restored proliferation, these effects were reversed by MPC inhibitor treatment (Supplementary Fig. 4g). Collectively, these results provide strong evidence supporting the specificity of *Mpc2* shRNA knockdown and functional relevance of MPC inhibition in modulating cell proliferation and pyruvate dependence in the context of Cpt1a deficiency. We appreciate the insightful comment, which has prompted us to conduct additional experiments to strengthen our data.

5. Fig. S6: Is it the author's model that increased oxidative stress induction contribute to decreased proliferation of Cpt1a KO cells? In this case does NAC rescue the proliferation of CPT1A KO cells? However, the authors also argue that Nrf2 activation downstream of oxidative stress induction is responsible for the compensatory increased glucose oxidation- in this case, would NAC block Nrf2 activation and further impair growth, similar to the Nrf2 inhibitors? How does NAC influence GLUT1 expression, glucose uptake, and glucose utilization for lactate vs. TCA cycle?

We thank the reviewer for their comment on Supplementary Figure 6 (now Supplementary Fig. 7), and for raising important questions regarding our model. Indeed, our model posits that increased oxidative stress induction contributes to the decreased proliferation observed in Cpt1a-deficient cells.

Regarding the role of NAC, it is important to note that several studies have demonstrated the ability of NAC to protect against oxidative stress by activating Nrf2, rather than inhibiting its function (9-11). However, the function of NAC on Nrf2 activity can be highly contradictory, as studies have shown that low doses of NAC promotes Nrf2 activation, while higher doses can drive Nrf2 inhibition (12). NAC serves as a precursor of cysteine, which is a critical component required for glutathione synthesis, which is an important antioxidant molecule required for maintaining cellular redox balance and is involved in the Nrf2 signaling pathway (13). Under conditions of oxidative stress, Keap1 undergoes modifications that prevent it from binding to Nrf2 (14). As a result, Nrf2 is stabilized and translocates to the nucleus, where it binds to antioxidant response elements (AREs), leading to their transcriptional activation (14). However, the relationship between Nrf2 activation and the expression of downstream target genes is complex. While Nrf2 activation typically leads to the upregulation of genes involved in antioxidant defense and detoxification pathways, this can be modulated in a context-dependent manner. In the case of NAC treatment, while it may promote Nrf2 activation initially, prolonged or excessive activation of Nrf2 signaling can lead to negative feedback regulation or crosstalk with other signaling pathways, resulting in the downregulation of certain Nrf2 target genes, such as Nqo1, Hmox1, Gclc, and G6pd (as observed in Supplementary Fig. 7f) (15). Additionally, NAC may exert its effects on other regulatory pathways or transcription factors that can influence the expression of antioxidant target genes independent of Nrf2 activity (16).

In line with these findings, our study shows that NAC partially rescues mitochondrial morphology and function (Supplementary Fig. 7, b-d) and can partially rescue the proliferation of Cpt1a-deficient cells (Supplementary Fig. 7e), similar to the effects observed with constitutive activation of Nrf2 by knockdown of its repressor, Keap1 (Fig. 5f). This supports the idea that alleviating oxidative stress in Cpt1a-deficient cells by NAC treatment can partially restore cellular proliferation. Notably, as Nrf2 and oxidative stress are not upregulated in wild-type cells, NAC does not significantly affect proliferation in these cells (Supplementary Fig. 7e). Furthermore, while some Nrf2 target gene are downregulated upon NAC treatment (Supplementary Fig. 7f), NAC can upregulate *Nrf2* itself, subsequently promoting *Glut1* expression and glucose uptake in cells lacking Cpt1a (Supplementary Fig. 7, g and h). Additionally, NAC promotes glucose utilization towards TCA cycle intermediates, favoring mitochondrial function over lactate production (Supplementary Fig. 7i). This recapitulates the phenotypes observed with Nrf2 activation by Keap1 knockdown, suggesting that Cpt1a-deficient cells rely on Nrf2 and glucose for oxidative phosphorylation (OXPHOS) and energy production. Moreover, excessive oxidative damage can indeed decrease proliferation, further supporting the role of Nrf2 in cellular homeostasis. We appreciate the thoughtful insights, which have helped refine our understanding of these complex mechanisms.

6. Fig. 5B: it is stated that wild type cells are not dependent on NRF2 for glucose uptake, and yet NRF2 inhibition does inhibit glucose uptake to some extent in wild-type cells.

We thank the reviewer for their insightful comment regarding the role of Nrf2 in glucose uptake in wild-type HER2+ cells. While it is indeed stated that wild-type cells are not solely dependent on Nrf2 for glucose uptake, it is important to note that NRF2 inhibition can still have an impact on glucose uptake in these cells, and we have ensured that we refrain from making this claim in the revised manuscript. Although wild-type HER2+ cells exhibit lower protein and gene expression levels of Nrf2 compared to Cpt1a-deficient cells (Fig. 4d and Supplementary Fig. 6b), Nrf2 is still present and can indirectly promote *Glut1* expression and facilitate glucose

uptake into these cell. Therefore, Nrf2 inhibition may partially impair Glut1 expression and glucose uptake in wild-type NIC cells, albeit not to the same extent as in Cpt1a-deficient cells. It is crucial to recognize that wild-type cells utilize multiple mechanisms, including Nrf2, to regulate glucose uptake, whereas Cpt1a-deficient cells upregulate and rely more heavily on Nrf2-mediated pathways to promote glucose uptake as their primary source of nutrients and for metabolic plasticity.

7. Fig. S8c: Why would i cells not respond to b-OHB when bOHB transport into the mitochondria and oxidation should not be dependent on CPT1a

We thank the reviewer for their question and observation. While it is true that the transport and oxidation of β -hydroxybutyrate (β OHB) within the mitochondria are not reliant on Cpt1a, it is important to consider the broader context of cellular metabolism and signaling pathways upon ablation of Cpt1a. Our research aims to identify alternative fuel sources that could bypass the loss of Cpt1a and long-chain fatty acid oxidation to uncover metabolic vulnerabilities. Although β OHB can indeed enter the mitochondria independent of Cpt1a, the overall metabolic state of the cell, including the availability of substrates and regulatory mechanisms, may influence its utilization.

Regarding the observation about β OHB ketone supplementation not affecting Cpt1a-deficient cell proliferation, it is important to note that this phenomenon can be attributed to several factors. Firstly, the expression of β -hydroxybutyrate dehydrogenase 1 (Bdh1), the enzyme responsible for metabolizing β OHB to acetoacetate, is significantly lower in Cpt1a-deficient cells (Supplementary Fig. 9d). As such, these cells may not be able to efficiently metabolize ketone bodies such as β OHB, irrespective of Cpt1a loss. Our findings indicate that β OHB supplementation and ketone oxidation cannot compensate for Cpt1a loss in HER2+ cells. It is plausible that diminished Cpt1a levels, resulting in decreased transport of long-chain fatty acids into the mitochondria, decrease the availability of acetyl CoA, which is a precursor for ketone body synthesis (17). Due to reduced levels of acetyl-CoA, the substrate availability for Bdh1 is reduced, potentially leading to decreased Bdh1 activity (Supplementary Fig. 9e) (17, 18). Additionally, decreased Cpt1a levels and impaired fatty acid oxidation, may lead to changes in the expression of transcription factors or metabolic regulators that regulate Bdh1 expression and activity (19).

Furthermore, we have observed that medium-chain fatty acids (MCFAs), namely caprylic acid, capric acid and lauric acid, can partially restore proliferation, migration, invasion and mitochondrial respiration, and are preferred lipid sources compared to ketones and long-chain fatty acids in Cpt1a-deficient NIC cells (Supplementary Fig. 11, a-c). These results indicate that MCFAs and glucose play a crucial role in sustaining cellular metabolism and more efficiently compensate for the energy demand in the absence of Cpt1a-mediated long-chain fatty acid entry into the mitochondria and oxidation. Similar observations have been noted previously in models of cardiac hypertrophy, which exhibit diminished Cpt1a activity and oxidation of long-chain fatty acids (20). In this model, glucose and short-chain fatty acids, rather than ketones, serve as an alternative fuel source to counter reduced long-chain fatty acid oxidation within the mitochondria and are capable of bypassing loss of Cpt1a to support energy production in pathologically failing hearts. We appreciate the insightful observation which have contributed significantly to our understanding of metabolic pathways and potential therapeutic targets in HER2+ breast cancer cells. Specifically, they bolster our rationale for incorporating the ketogenic diet supplemented with long and medium-chain fatty acids to reinforce our conclusions.

8. Similar to point 3 above, would formulating a ketogenic diet with medium chain triglycerides rescue the effects of the KD + CPT1A KO combination?

We thank the reviewer for their suggestion. As previously indicated, supplementation with medium-chain fatty acids (MCFAs) can partially rescue proliferation, migration, invasion, OCR, and glucose dependence, effectively bypassing Cpt1a loss in NIC cells (Supplementary Fig. 11, a-e). These data argue that Cpt1a-deficient cells are capable of utilizing MCFAs as alternative fuel sources. To assess whether these findings could be translated *in vivo*, we formulated a ketogenic diet with MCFAs, termed the Medium-Chain Ketogenic Diet (MCKD). This diet replaced the lipid composition from long-chain FAs to MCFAs, primarily composed of coconut oil, and the remaining diet was the same as the Long-Chain Ketogenic Diet (LCKD) used previously

(Supplementary 10a and Supplementary Table S2). In mice bearing NIC/Cpt1a^{+/+} and NIC/Cpt1a^{L/L} tumors, the MCKD successfully induced ketosis, as evidenced by reduced plasma glucose, increased 3-OHB levels, and decreased total body weight compared to the control diet, similar to observations with the LCKD in Fig. 6 and Supplementary Fig. 10 (Supplementary Fig. 11, f and g). Although the MCKD was able to recapitulate the increase in tumor volume and decrease animal survival observed in wild-type ErbB2+ tumors on the LCKD, the impact on Cpt1a-deficient ErbB2+ tumors was negligible (Supplementary Fig. 11, h and i). Conversely, combining Cpt1a ablation with the LCKD markedly decreased tumor growth and enhanced survival compared to all other conditions (Fig. 6, d-f). Taken together, our findings indicate that supplementation with MCFAs in the ketogenic diet cannot bypass and rescue ErbB2+ tumor growth and proliferation upon Cpt1a ablation. Therefore, combining a ketogenic diet composed of long-chain rather than MCFAs with FAO blockade, through Cpt1a targeting, proves to be an effective therapeutic strategy.

9. Fig. 7D-F: There does not appear to be a synergy between CPT1A loss and HER2 inhibition- while the combination does lead to the slowest growing tumors, HER2 inhibition seems to have a stronger relative effect in WT tumors compared to CPT1A KO tumors. The combination is likely more additive than synergistic.

We thank the reviewer for this valuable feedback. Regarding the lack of synergy between Cpt1a loss and HER2 inhibition in our study, we agree with your assessment that the combination effect appears to be more additive than synergistic in HER2+ breast cancer cells. We apologize for any confusion our previous wording may have caused, and have promptly made the necessary revisions to our manuscript to accurately reflect this finding. We thank the reviewer for these comments, which have allowed us to reinforce the accuracy and conclusion of our study.

10. Abstract states that “combining a ketogenic diet with an anti-ErbB2 monoclonal antibody in the context of CPT1A deficiency significantly reduced tumor growth and increased survival”. Unless I missed it, where is this data?

We apologize for the confusion and oversight regarding the abstract, and you are correct in pointing out this discrepancy. Our intention was to convey that combining either a ketogenic diet, composed of long-chain triglycerides, or an anti-ErbB2 monoclonal antibody, with Cpt1a inhibition can significantly reduce tumor growth and increase animal survival. However, it seems there was confusion in our wording as the combination of all three treatment arms was not conducted. Instead, individual combination strategies were explored to assess alternate candidate approaches. As such, we have rectified this error in the abstract to accurately reflect our findings. We thank the reviewer for bringing this to our attention.

Reviewer #3:

In this report, Muller and colleagues carry out an analysis of the effects of loss of Cpt1a in HER2+ mammary tumors. In addition to examining growth, angiogenesis, and metastasis, they carry out an extensive analysis of the metabolic consequences of Cpt1a knockout. These studies not only revealed impaired mitochondrial function associated with an increased reliance on glycolysis and the TCA cycle for energy production, but also revealed a significant upregulation of oxidative stress and adaptive upregulation of the anti-oxidant program regulated by NRF2. The metabolic studies were quite thorough with metabolic tracing studies as well as analysis of effects of inhibitors to counteract adaptive programs. In addition, the authors explored several candidate therapeutic approaches and identified some candidate combinations. This report was impressive in the breadth and depth of analyses as well as inclusion of appropriate replicates and alternate approaches to investigate their findings.

These studies provide new insights into metabolic programs associated perturbation of fatty acid oxidation in HER2 tumors, and reveal candidate therapeutic strategies to target this pathway and adaptive programs that result from its inhibition.

There were only two minor issues that would improve the manuscript if addressed:

1) Figure 6d and e: In Figure 6d, the data shows essentially no difference in weight of the HER2 tumors with or without Cpt1a KO at tumor endpoint; however figure 6e, which shows tumor volume, show ~ a 7- to 10- fold difference in tumor volume. Similarly, the Keto diet Cpt1a KO mice display a dramatic difference in tumor

volume relative to the control WT mice. It is likely that this is an explanation for this but it isn't clear here and it is important to address this.

We apologize for the confusion and wish to clarify that the observed difference stems from the method of data collection of tumour volume and tumour weight. Specifically, the tumour volume graph (Fig. 6e) depicts measurements taken over a 10-week period from palpable tumours in live mice, whereas tumour weight data (Fig. 6d) were acquired at tumour endpoint, defined as when tumour volume reaches 2.5 cm³ and the animal was euthanized. The 10-week duration depicted in Fig. 6e was chosen based on the endpoint typically observed for wild-type tumours under normal chow conditions. However, this timeframe is not adequate for Cpt1a-deficient tumours. Cpt1a-deficient tumours tend to reach their endpoint approximately at 19 weeks for mice on a normal diet and at 28 weeks for those on a keto diet, and even at 28 weeks, some tumors do not fully reach the endpoint (Fig. 6f). While the 10-week representation in the tumour volume curve (Fig. 6d) offers an overview of the phenotype, additional curves indicating the endpoint for Cpt1a-deficient tumours on normal and ketogenic chow have also been provided (Supplementary Fig. 10c). Due to limitations in our Animal Use Protocol, obtaining weekly tumour weight measurements, without sacrificing the mouse, is not feasible. We apologize for any confusion and have enhanced the clarity of our methods for tumour volume and weight collection in both the Methods section and the figure legend.

2) A keto diet was shown to increase tumor size and decrease survival in mice bearing wild-type ErbB2+ tumors, whereas “combining Cpt1a ablation with the ketogenic diet markedly decreased tumor growth and enhanced survival compared to all other conditions”. This is an interesting finding which provided useful information to the authors comprehensive analysis of Cpt1a function in HER2 tumors; however they also discuss use of ketogenic diets together with CPT1A inhibition as therapeutic strategy. It is recommended to add a cautionary note about the use of any strategy in which one arm of combination therapy enhances tumor growth.

We thank the reviewer for this suggestion and appreciate the insightful observation regarding the potential implications of therapies that may inadvertently enhance tumor growth. While our study highlights the promising therapeutic potential of combining Cpt1a ablation with a ketogenic diet in the context of ErbB2+ tumors, it is crucial to acknowledge the complexity of such combination strategies. Specifically, caution should be exercised when considering therapeutic approaches in which one arm of the combination, the ketogenic diet, may promote tumor growth when used alone. We have made appropriate changes in the text to reflect this comment. We thank the Reviewer for this suggestion, which we believe have strengthened the conclusions of the paper. We have duly noted this concern and recognize the importance of addressing it within the context of our findings.

3) One question that comes from this analysis is how glutamine metabolism was affected by Cpt1a deletion. It isn't critical for the report which is quite thorough, but if the authors looked at this in any of their metabolic studies, it would be informative to add this.

We thank the Reviewer for this insightful comment. We have indeed explored the impact of Cpt1a deletion on glutamine metabolism in HER2+ breast cancer cells. Our investigation included a comprehensive analysis similar to that depicted in Supplementary Figure 9a. Notably, our findings revealed that restricting glutamine availability in the media did not significantly reduce the proliferation of both Cpt1a-proficient and -deficient HER2+ breast tumor cells independent of glutamine concentration (Rebuttal Fig. 1a). This indicates that HER2+ cells, with and without Cpt1a, do not exhibit heightened sensitivity to low glutamine conditions. Furthermore, gene expression analysis of enzymes involved in glutamine transport (*Slc1a5*), catabolism (*Idh2*, *Got1*, *Got2*, and *Glud1*), and synthesis (*Gls1* and *Gls2*) were unchanged Cpt1a-deficient HER2+ cells compared to wild-type cells (Rebuttal Fig. 1b) indicating that while HER2+ cells can utilize and metabolize glutamine, it is not their primary source of nutrients. Instead, our data support the notion that HER2+ rely predominantly on fatty acids and glucose as their primary sources of energy. We thank the Reviewer for these helpful suggestions.

Rebuttal Fig. 1

In summary, we sincerely thank all the reviewers for taking the time to read our manuscript and provide constructive feedback, and we are very pleased to present this significantly strengthened manuscript for your consideration. We trust that you will find the revised manuscript suitable for publication in Nature Communications.

Yours sincerely,

Professor William J. Muller
 Goodman Cancer Institute and Department of Biochemistry
 McGill University
 Montreal, Quebec, Canada

References

1. Dubikovskaya E, Chudnovskiy R, Karateev G, Park HM, and Stahl A. Measurement of long-chain fatty acid uptake into adipocytes. *Methods Enzymol.* 2014;538:107-34.
2. Stahl A, Evans JG, Pattel S, Hirsch D, and Lodish HF. Insulin causes fatty acid transport protein translocation and enhanced fatty acid uptake in adipocytes. *Dev Cell.* 2002;2(4):477-88.
3. Doege H, Grimm D, Falcon A, Tsang B, Storm TA, Xu H, et al. Silencing of hepatic fatty acid transporter protein 5 in vivo reverses diet-induced non-alcoholic fatty liver disease and improves hyperglycemia. *J Biol Chem.* 2008;283(32):22186-92.
4. Hirsch D, Stahl A, and Lodish HF. A family of fatty acid transporters conserved from mycobacterium to man. *Proceedings of the National Academy of Sciences of the United States of America.* 1998;95(15):8625-9.
5. Stahl A, Hirsch DJ, Gimeno RE, Punreddy S, Ge P, Watson N, et al. Identification of the major intestinal fatty acid transport protein. *Mol Cell.* 1999;4(3):299-308.
6. Tan Z, Xiao L, Tang M, Bai F, Li J, Li L, et al. Targeting CPT1A-mediated fatty acid oxidation sensitizes nasopharyngeal carcinoma to radiation therapy. *Theranostics.* 2018;8(9):2329-47.
7. Sheppard S, Srpan K, Lin W, Lee M, Delconte RB, Owyong M, et al. Fatty acid oxidation fuels natural killer cell responses against infection and cancer. *Proceedings of the National Academy of Sciences of the United States of America.* 2024;121(11):e2319254121.
8. Jadhav HB, and Annapure US. Triglycerides of medium-chain fatty acids: a concise review. *J Food Sci Technol.* 2023;60(8):2143-52.
9. Jannatifar R, Parivar K, Hayati Roodbari N, and Nasr-Esfahani MH. The Effect of N-Acetyl-Cysteine on NRF2 Antioxidant Gene Expression in Asthenoteratozoospermia Men: A Clinical Trial Study. *Int J Fertil Steril.* 2020;14(3):171-5.
10. Crinelli R, Zara C, Galluzzi L, Buffi G, Ceccarini C, Smietana M, et al. Activation of NRF2 and ATF4 Signaling by the Pro-Glutathione Molecule I-152, a Co-Drug of N-Acetyl-Cysteine and Cysteamine. *Antioxidants (Basel).* 2021;10(2).
11. Wolfram T, Schwarz M, Reuß M, Lossow K, Ost M, Klaus S, et al. N-Acetylcysteine as Modulator of the Essential Trace Elements Copper and Zinc. *Antioxidants (Basel).* 2020;9(11).
12. Yu Z, Yu K, Wu S, Zhao Q, Guo Y, Liu H, et al. Two contradictory facades of N-acetylcysteine activity towards renal carcinoma cells. *Journal of Taibah University for Science.* 2022;16(1):423-31.
13. Sahasrabudhe SA, Terluk MR, and Kartha RV. N-acetylcysteine Pharmacology and Applications in Rare Diseases—Repurposing an Old Antioxidant. *Antioxidants (Basel).* 2023;12(7).
14. Ma Q. Role of nrf2 in oxidative stress and toxicity. *Annu Rev Pharmacol Toxicol.* 2013;53:401-26.
15. Liu S, Pi J, and Zhang Q. Signal amplification in the KEAP1-NRF2-ARE antioxidant response pathway. *Redox Biology.* 2022;54:102389.
16. Sahasrabudhe SA, Terluk MR, and Kartha RV. N-acetylcysteine Pharmacology and Applications in Rare Diseases—Repurposing an Old Antioxidant. *Antioxidants.* 2023;12(7):1316.
17. Hwang CY, Choe W, Yoon KS, Ha J, Kim SS, Yeo EJ, et al. Molecular Mechanisms for Ketone Body Metabolism, Signaling Functions, and Therapeutic Potential in Cancer. *Nutrients.* 2022;14(22).
18. Stagg DB, Gillingham JR, Nelson AB, Lengfeld JE, d'Avignon DA, Puchalska P, et al. Diminished ketone interconversion, hepatic TCA cycle flux, and glucose production in D-β-hydroxybutyrate dehydrogenase hepatocyte-deficient mice. *Mol Metab.* 2021;53:101269.
19. Nakagawa Y, Satoh A, Tezuka H, Han SI, Takei K, Iwasaki H, et al. CREB3L3 controls fatty acid oxidation and ketogenesis in synergy with PPARα. *Sci Rep.* 2016;6:39182.
20. Carley AN, Maurya SK, Fasano M, Wang Y, Selzman CH, Drakos SG, et al. Short-Chain Fatty Acids Outpace Ketone Oxidation in the Failing Heart. *Circulation.* 2021;143(18):1797-808.

REVIEWERS' COMMENTS

Reviewer #2 (Remarks to the Author):

Thank you for extensively addressing my comments. The revision has significantly strengthened the manuscript, and the authors have sufficiently addressed my comments.

Reviewer #3 (Remarks to the Author):

The authors has address the issues that I raised in the initial review.

William J. Muller, Ph.D., *FRSC*
Rosalind and Morris Goodman Cancer Institute
Cancer Pavilion, McGill University
1160 Pine Ave. West, Room 516
Montreal, Québec, Canada H3A 1A3

July 12th, 2024

Response to Reviewer's Comments

We are very pleased to resubmit our revised manuscript "**Targeting Fatty Acid Oxidation Enhances Response to HER2-targeted Therapy**" (NCOMMS-24-02043-B) to Nature Communications. We are grateful for the positive responses to our work, and we believe that the revised manuscript we submit here has been improved further and is now suitable for publication in Nature Communications. We have followed the author checklist for revised submission to ensure that our manuscript meets the standards of Nature Communication for publication. Please find below our responses to the reviewers' comments, which are also included in italics for ease of reference. Other changes to the text were made for clarity and in adherence with Nature Communications editorial suggestions.

Reviewer 2:

Thank you for extensively addressing my comments. The revision has significantly strengthened the manuscript, and the authors have sufficiently addressed my comments.

We sincerely thank the reviewer for their comments and suggestions, which have allowed us to reinforce the conclusions of our study.

Reviewer 3:

The authors has address the issues that I raised in the initial review.

We thank the reviewer for their insightful comments, suggestions and time in reviewing our manuscript.

In summary, we sincerely thank you and the reviewers for taking the time to read our manuscript and provide constructive feedback, and we are very pleased to present this significantly strengthened manuscript for your consideration. We trust that you will find the revised manuscript suitable for publication and we look forward to hearing from you again soon.

Yours sincerely,

Professor William J. Muller
Rosalind and Morris Goodman Cancer Institute and Department of Biochemistry
McGill University
Montreal, Quebec, Canada